# Analysis of combinatorial chemokine receptor expression dynamics using multi-receptor reporter mice

Laura Medina-Ruiz[1]*, Robin Bartolini[1], Gillian J Wilson[1], Douglas P Dyer[1†], Francesca Vidler[1,2], Catherine E Hughes[1], Fabian Schuette[1], Samantha Love[1], Marieke Pingen[1], Alan James Hayes[1], Jun Fu[2,3], Adrian Francis Stewart[2,4], Gerard J Graham[1]*

[1]Chemokine Research Group, Institute of Infection, Immunity and Inflammation, College of Medical, Veterinary and Life Sciences, University of Glasgow, Glasgow, United Kingdom; [2]Center for Molecular and Cellular Bioengineering, Biotechnology Center, Technische Universität Dresden, Dresden, Germany; [3]Shandong University–Helmholtz Institute of Biotechnology, State Key Laboratory of Microbial Technology, School of Life Science, Shandong University, Shandong, China; [4]Max-Planck-Institute for Cell Biology and Genetics, Dresden, Germany

*For correspondence:
Laura.medina-ruiz@glasgow.ac.
uk (LM-R);
gerard.graham@glasgow.ac.uk
(GJG)

Present address: †Wellcome
Centre for Cell-Matrix Research
and Lydia Becker Institute of
Immunology and Inflammation,
Faculty of Biology, Medicine and
Health, Manchester Academic
Health Science Centre, University
of Manchester, Manchester,
United Kingdom

**Competing interest:** The authors
declare that no competing
interests exist.

**Reviewing Editor:** Florent
Ginhoux, Agency for Science
Technology and Research,
Singapore

**Abstract** Inflammatory chemokines and their receptors are central to the development of inflammatory/immune pathologies. The apparent complexity of this system, coupled with lack of appropriate in vivo models, has limited our understanding of how chemokines orchestrate inflammatory responses and has hampered attempts at targeting this system in inflammatory disease. Novel approaches are therefore needed to provide crucial biological, and therapeutic, insights into the chemokine-chemokine receptor family. Here, we report the generation of transgenic multi-chemokine receptor reporter mice in which spectrally distinct fluorescent reporters mark expression of CCRs 1, 2, 3, and 5, key receptors for myeloid cell recruitment in inflammation. Analysis of these animals has allowed us to define, for the first time, individual and combinatorial receptor expression patterns on myeloid cells in resting and inflamed conditions. Our results demonstrate that chemokine receptor expression is highly specific, and more selective than previously anticipated.

## Editor's evaluation

The manuscript describes the potential of unique multi-chemokine receptor reporter mice to track CCR expression dynamics in myeloid cells in steady state and inflammatory conditions which may overcome some existing limitations to finding accurate antibodies for these receptors. This paper will be of broad interest to investigators studying the role of chemokines and their receptors in animal models of human disease. The conclusions of the authors are generally well-supported by the data and provide novel and important insights into the hierarchy of chemokine receptor expression during the recruitment of myeloid cells to sites of inflammation.

## Introduction

Chemokines, and their receptors, are primary regulators of in vivo leukocyte migration and central orchestrators of innate and adaptive immune responses (*Griffith et al., 2014*; *Rot and von Andrian, 2004*). Chemokines are defined by a conserved cysteine motif and subdivided into CC, CXC, CX$_3$C, and XC subfamilies according to the specific configuration of this motif (*Bachelerie et al., 2014*;

*Rot and von Andrian, 2004*). Chemokines signal through seven-transmembrane spanning G protein-coupled receptors expressed by immune and inflammatory cells (*Bachelerie et al., 2014*; *Rot and von Andrian, 2004*), and these receptors are named according to the subfamily of chemokines with which they interact (CCR, CXCR, XCR, and CX$_3$CR).

Chemokines and their receptors can be functionally classified as either homeostatic or inflammatory according to the in vivo contexts in which they function (*Mantovani, 1999*; *Zlotnik and Yoshie, 2000*). Inflammatory chemokines and their receptors mediate leukocyte recruitment to inflamed, damaged, or infected sites and are prominent contributors to a large number of autoimmune and inflammatory diseases (*Viola and Luster, 2008*). Chemokine receptors therefore represent important therapeutic targets (*Proudfoot et al., 2015*). However, to date, only two chemokine receptor antagonists have been approved for therapeutic use (Plerixafor, targeting CXCR4, and Maraviroc, targeting CCR5), and no antagonists have yet been approved for treating immune/inflammatory disease (*Bachelerie et al., 2014*). There are a number of explanations for this failure (*Schall and Proudfoot, 2011*), prominent amongst which is the fact that inflammatory chemokine and chemokine receptor biology is highly complex. For example, multiple chemokines are simultaneously expressed at inflamed sites and these interact in a complex, and poorly understood, manner with different inflammatory chemokine receptors (*Bachelerie et al., 2014*). To complicate matters further, reports in the literature indicate that distinct leukocyte subsets simultaneously express multiple chemokine receptors (*Bachelerie et al., 2014*; *Griffith et al., 2014*). As a result of this complexity it is currently difficult to precisely define in vivo roles for inflammatory chemokine receptors and we therefore lack a clear understanding of their integrated involvement in the orchestration of the inflammatory response.

The current study focuses on four inflammatory CC chemokine receptors (iCCRs) (*Dyer et al., 2019*): CCR1, CCR2, CCR3, and CCR5, which exemplify the ligand-receptor interaction complexity noted above. Collectively, these receptors direct non-neutrophilic myeloid cell trafficking at rest and during inflammation (*Pease, 2011*; *Shi and Pamer, 2011*) and are key players in inflammatory diseases. However, the resting and inflamed expression of these receptors on individual leukocyte populations has not been clearly defined. In addition, their individual roles in leukocyte mobilisation, recruitment, and intra-tissue dynamics are still unclear. The use of knockout mice to study their function has been confused by partial phenotypes and conflicting results (*Bennett et al., 2007*; *Humbles et al., 2002*; *Ma et al., 2002*; *Pope et al., 2005*; *Rottman et al., 2000*; *Tran et al., 2000*), leading to suggestions of redundancy in their function. There is therefore a pressing need to develop powerful novel tools to precisely define the temporo-spatial patterns of expression of these receptors at rest and during inflammation and in so doing to precisely delineate their roles in the physiological and pathological inflammatory response.

Here, we report the generation of an iCCR reporter (iCCR-REP) mouse strain expressing spectrally distinct fluorescent reporters for CCR1, CCR2, CCR3, and CCR5. We have used these mice to provide, for the first time, a comprehensive analysis of iCCR expression on bone marrow (BM), peripheral blood, and tissue-resident myeloid cells at rest and during inflammatory responses. In contrast to published data, our analysis indicates selective receptor expression in individual cell types, in resting and acute inflammatory contexts and suggests little, if any, redundancy in function. We propose that these mice represent a transformational addition to the suite of mouse models available for analysing inflammatory chemokine receptor function in vivo and that they will be instrumental in helping to deconvolute the complexity of the chemokine-driven inflammatory response in a variety of pathological contexts.

## Results
### Generation of iCCR-reporter mice

The inflammatory *Ccr* genes are organised in a single 170 kb genomic cluster, which contains no other genes and which is highly conserved among mammals (*Nomiyama et al., 2011*) and located on mouse chromosome 9. To produce the iCCR-reporter (iCCR-REP) mice, we generated a recombineered version of a bacterial artificial chromosome (BAC) encompassing the cluster (inflammatory *Ccr* gene-REP cluster), in which the coding sequence of each of the inflammatory *Ccr* genes was replaced with sequences encoding spectrally distinct fluorescent proteins. The reporters used in this study [mTagBFP2 (*Subach et al., 2011*), Clover, mRuby2 (*Lam et al., 2012*), and iRFP682 (*Shcherbakova*

and Verkhusha, 2013)] were selected on the basis of their discrete excitation and emission spectra (*Figure 1A*). Using counterselection recombineering (*Wang et al., 2014*), *Ccr1* was replaced with Clover, *Ccr2* with mRuby2, *Ccr3* with mTagBFP2, and *Ccr5* with iRFP682 (*Figure 1Bi and ii*). Transgenic iCCR-REP mice were generated by pro-nuclear injection of the inflammatory *Ccr* gene-REP BAC (*Figure 1Biii*). Using targeted locus amplification (TLA) (*de Vree et al., 2014*), the inflammatory *Ccr* gene-REP cluster was located to chromosome 16:82389380–82392016 (*Figure 1Ci*), where 5–8 copies of the BAC were inserted in a head-tail manner. Insertion of the inflammatory *Ccr* gene-REP clusters lead to the deletion of a 2.5 kb genomic region that contained no coding or regulatory sequences (*Figure 1Cii*).

Thus, using recombineering, we have generated transgenic (iCCR-REP) mice expressing spectrally distinct reporters for each of the iCCRs.

The iCCR-REP mice maintain the original inflammatory *Ccr* gene cluster on chromosome 9. To confirm that the transgene does not interfere with normal iCCR-dependent myeloid cell recruitment, we examined myeloid cell population sizes in different tissues by flow cytometry (gating strategies: *Figure 1—figure supplements 1 and 2*). Analysis of resting mice demonstrated that the myeloid cell content in these tissues was indistinguishable between iCCR-REP and wild type (WT) animals for all populations analysed (*Figure 1—figure supplement 3*). Next, we tested for possible effects of the transgene on myeloid cell recruitment to inflamed sites. To this end we used two different models. First, we used the air-pouch model of inflammation, involving the generation of an air cavity under the dorsal skin of the mouse and the injection of carrageenan into the cavity to induce inflammation. In line with the results obtained from resting tissues, analysis of the myeloid cell content in the membrane surrounding the inflamed air-pouch, as well as in the fluid collected from the inflamed cavity, showed no differences between iCCR-REP and WT animals (*Figure 1D*). We also analysed cellular content in inflamed lungs of mice that received an intranasal dose of PBS (control) or lipopolysaccharide (LPS). The data generated demonstrate a reduction in the relative percentage of monocytes, macrophages, and neutrophils in the LPS-treated lungs, along with an expected reduction in relative percentages of alveolar macrophages and dendritic cells (*Figure 1E*). In keeping with previous reports (*Righetti et al., 2018*), eosinophils were not recruited to the lung in response to acute LPS. Again, no differences were observed between iCCR-REP and WT animals in the relative percentage of any of the populations measured in either control, or LPS-treated, lungs.

Together, these data demonstrate that iCCR-REP mice have normal iCCR-dependent myeloid cell recruitment/migration dynamics at rest and under inflammatory conditions.

## iCCR-reporter expression accurately replicates iCCR surface presentation

To confirm that reporter expression faithfully recapitulates iCCR surface presentation, we compared iCCR antibody binding and iCCR reporter expression by flow cytometry (gating strategies: *Figure 1—figure supplements 1A and 2B*). As shown in *Figure 2A*, the majority of splenic Ly6C$^{hi}$ monocytes from resting WT mice displayed anti-CCR2 antibody staining and essentially all mRuby2/CCR2 expressing Ly6C$^{hi}$ monocytes in iCCR-REP mice were co-positive for CCR2 antibody staining. Background autofluorescence was undetectable in WT or iCCR-deficient (iCCR-def) mice (these mice are homozygous for a deletion encompassing the entire iCCR locus; *Dyer et al., 2019*) although we routinely detected non-specific antibody staining on iCCR-def cells. Similar results were obtained when analysing splenic SiglecF$^+$ eosinophils for expression of mTagBFP2/CCR3. Splenic iCCR-REP eosinophils expressing mTagBFP2 simultaneously displayed surface CCR3 antibody staining (*Figure 2B*). CCR1 and CCR5 were difficult to detect in splenocytes and were therefore analysed in the kidney in which tissue both reporters were expressed on CD11b$^+$F480$^+$ macrophages. iCCR-REP macrophages expressing iRFP682 simultaneously showed surface CCR5 antibody staining (*Figure 2C*) and low-level non-specific antibody staining was observed on iCCR-def cells. Clover/CCR1 was detected on kidney CD11b$^+$F480$^+$ macrophages of iCCR-REP mice (*Figure 2Di and ii*). However, in our hands, no commercially available antibodies were able to detect CCR1 on any tissues analysed. For that reason, we used RNAscope to confirm expression of CCR1 on kidney cells. As shown in *Figure 2Diii and iv*, CCR1 transcripts were clearly detectable on WT and iCCR-REP, but not iCCR-def kidney sections.

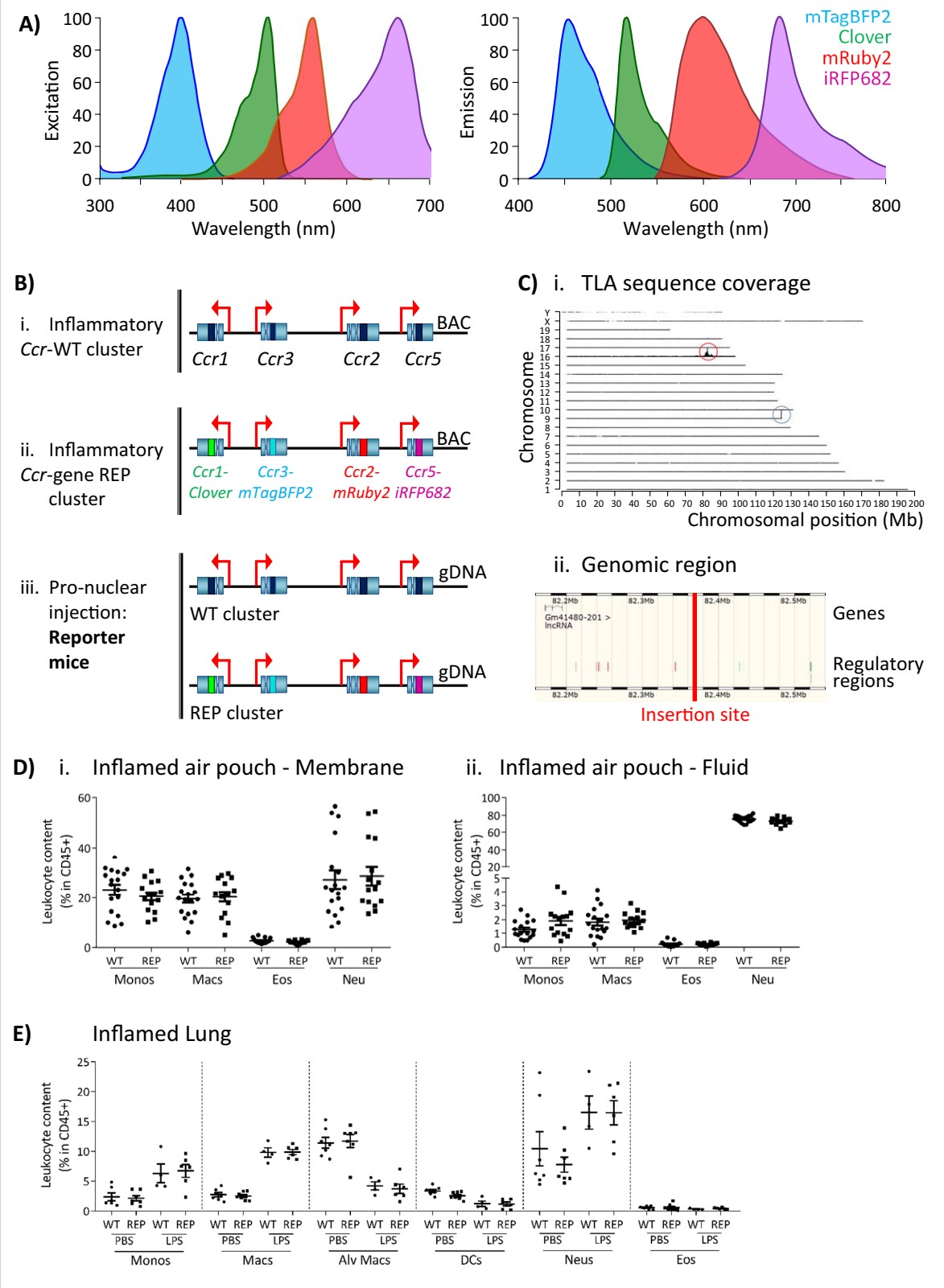

**Figure 1.** Generation of the reporter mice. (**A**) Reporters were selected for this study based on their discrete excitation and emission spectra. (**B**) (i) The inflammatory *Ccr* genes were targeted in a bacterial artificial chromosome (BAC). (ii) The coding sequence of each inflammatory *Ccr* was replaced with a different fluorescent reporter (ii). (iii) Pro-nuclear injection was then used to generate the transgenic reporter mice. (**C**) (**i**) The inflammatory *Ccr* gene-REP cluster inserted into chromosome 16 (red circle), as determined by targeted locus amplification (TLA). The blue circle represents the endogenous locus.

*Figure 1 continued on next page*

*Figure 1 continued*

(ii) The insertion site does not contain any other genes or regulatory regions. (**D**) Leukocyte counts determined by flow cytometry in the inflamed air-pouch. Data are shown for (**i**) the membrane that surrounds the air-pouch and for (**ii**) the lavage fluid. (**E**) Leukocyte counts determined by flow cytometry in PBS- or LPS-treted lungs. Data on D and E are shown as mean ± SEM and are compiled from at least two separate experiments. Normally distributed data were analysed using unpaired t-test with or without Welch's correction, according to their standard deviations. Not normally distributed data were analysed using Mann-Whitney or Kolmogorov-Smirnov, according to their standard deviations. Each data point represents a measurement from a single mouse (N=15-18 in D; N=4-6 in E). Abbreviations are: Monos, monocytes; Macs, macrophages; Neu, neutrophils; Eos, eosinophils; DCs, dendritic cells; Alv macs, alveolar macrophages. See also *Figure 1—figure supplement 3*.

The online version of this article includes the following figure supplement(s) for figure 1:

**Figure supplement 1.** Gating strategies used in the study.

**Figure supplement 2.** Gating strategies used in the study.

**Figure supplement 3.** Normal leukocyte recruitment in the reporter mice.

Thus, these data confirm that expression of the reporters in the iCCR-REP mice faithfully reflects iCCR expression and surface presentation. These mice, therefore, represent a unique, and validated, resource to examine individual and combinatorial iCCR expression in leukocytes.

## BM and circulating leukocytes express specific iCCRs

Next, we used flow cytometric analysis to examine iCCR expression in the iCCR-REP mice with WT littermates as controls for background autofluorescence. We first determined iCCR reporter expression in myeloid cells from resting BM and blood. Cell suspensions were prepared from both compartments and stained with leukocyte subset-specific antibodies (gating strategies: *Figure 1—figure supplement 1B*). We initially assessed iCCR reporter expression in Ly6C$^{hi}$ monocytes. As expected, and in agreement with previous reports (*Geissmann et al., 2003*; *Saederup et al., 2010*), the majority of Ly6C$^{hi}$ monocytes were positive for mRuby2/CCR2, in BM and in blood. mTagBFP2/CCR3 and iRFP682/CCR5 were not detected on monocytes. However, Clover/CCR1 was seen on a small number of Ly6C$^{hi}$ monocytes, representing approximately 3% of the population (*Figure 3Ai-iv*). Interestingly, in all cases, Clover was co-expressed with mRuby2 (*Figure 3Av-vii*).

SiglecF$^{+}$ eosinophils exclusively expressed mTagBFP2/CCR3 (*Figure 3Bi*) from the reporter cluster. However, only approximately 50% of the population expressed this reporter in the BM (*Figure 3Bii*), while levels increased as cells entered the circulation, where the majority of eosinophils now expressed mTagBFP2 (*Figure 3Biii*). We did not detect expression of any of the iCCR reporters in resting Ly6G$^{+}$ neutrophils (data not shown).

Thus, the iCCR-REP mice demonstrate that, with the exception of a small population of monocytic cells, individual iCCRs display selective association with discrete myeloid lineages in BM and peripheral blood.

## iCCR expression in resting tissues

Expression of the iCCR reporters was next assessed in resident myeloid cell populations of resting lungs and kidneys (gating strategies: *Figure 1—figure supplement 2*). The majority of Ly6C$^{hi}$ monocytes from resting lungs expressed mRuby2/CCR2, with only a very small fraction co-expressing Clover/CCR1 or iRFP682/CCR5 (*Figure 4Ai–iii* and *Figure 4—figure supplement 1*). In contrast, a high proportion of the monocyte-derived CD11b$^{+}$F480$^{+}$MHCII$^{Lo}$ interstitial macrophages (IMs) (IMs defined as in *Chakarov et al., 2019*; *Gibbings et al., 2017*; *Schyns et al., 2019*) expressed Clover/CCR1 and iRFP682/CCR5, with only a small fraction expressing mRuby2/CCR2 (*Figure 4Bi-iv*). Interestingly, in this case, Clover/CCR1 and iRFP682/CCR5 are mainly expressed independently of mRuby2/CCR2 (*Figure 4Bv* and *Figure 4—figure supplement 1*), suggesting that macrophages downregulate CCR2 and induce CCR1 and CCR5 as they differentiate from infiltrating monocytes. In line with these findings, we observed that the mean fluorescence intensity (MFI) of mRuby2 in CCR2$^{+}$ IMs was lower than that in mRuby2/CCR2$^{+}$ monocytes (*Figure 4Bvi*). This confirms that the CCR2 downregulation does not simply reflect reduction in the number of macrophages expressing the receptor but also reduced expression.

Similar results were obtained from resting kidneys. Again, Ly6C$^{hi}$ monocytes expressed almost exclusively mRuby2/CCR2, with only some co-expressing Clover/CCR1 or iRFP682/CCR5 (*Figure 5Ai–iii*

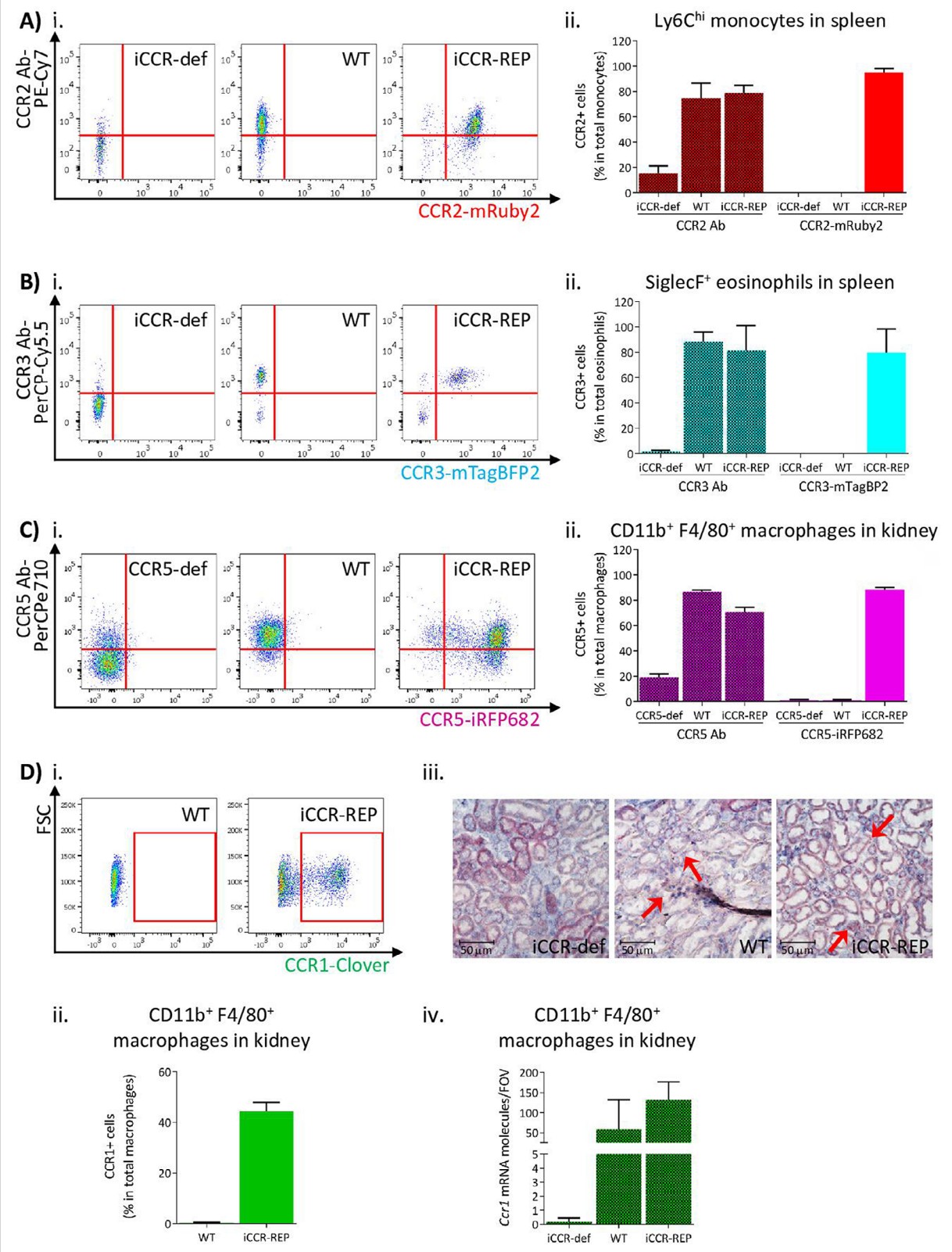

**Figure 2.** Inflammatory CC chemokine receptor (iCCR) reporter expression accurately mirrors iCCR surface presentation. (**A**) (i) Flow cytometric analysis of CCR2 antibody binding and mRuby2 expression in spleen Ly6C$^{hi}$ monocytes at rest. (ii) Quantification of the percentage of Ly6C$^{hi}$ monocytes binding CCR2 antibody and expressing mRuby2. (**B**) (i) Flow cytometric analysis of CCR3 antibody binding and mTagBFP2 expression in spleen SiglecF$^{+}$ eosinophils at rest. (ii) Quantification of the percentage of SiglecF$^{+}$ eosinophils binding CCR3 antibody and expressing mTagBFP2. (**C**) (i) Flow

*Figure 2 continued on next page*

*Figure 2 continued*

cytometric analysis of CCR5 antibody binding and iRFP682 expression in kidney CD11b⁺F4/80⁺ macrophages at rest. (ii) Quantification of the percentage of CD11b⁺F4/80⁺ macrophages binding CCR5 antibody and expressing iRFP682. (**D**) (i) Flow cytometric analysis of Clover expression in kidney CD11b⁺F4/80⁺ macrophages at rest. (ii) Quantification of the percentage of CD11b⁺F4/80⁺ macrophages expressing Clover. (iii) Brightfield images of resting kidneys showing CCR1 mRNA molecules detected by RNAscope analysis in the form of red precipitate dots (arrows). (iv) CCR1 mRNA molecule counts per field of view. Data information: data on Aii, Bii, Cii, Dii, and Div are shown as mean ± SD and are compiled from at least two separate experiments (N=4-7 in A, B and C: N=2-3 in D).

and *Figure 5—figure supplement 1*). However, monocyte-derived CD11b⁺F480⁺ macrophages (*Puranik et al., 2018*) mainly expressed Clover/CCR1 and iRFP682/CCR5, with only a small fraction retaining mRuby2/CCR2 expression (*Figure 5Bi–v* and *Figure 5—figure supplement 1*). Consistent with the observations in lung, the mRuby2/CCR2⁺ macrophages expressed lower mRuby2/CCR2 levels than Ly6Cʰⁱ monocytes as confirmed by the lower MFI for mRuby2 in this population (*Figure 5Bvi*). Together, these results suggest that this iCCR reporter expression pattern is consistent across different tissues under resting conditions.

We next assessed iCCR reporter expression in lung CD11b⁺F480⁺MHCIIʰⁱ IMs (*Chakarov et al., 2019*). As shown in *Figure 4—figure supplement 2*, F480ᴸᵒMHCIIʰⁱ IMs expressed reporters for CCRs 1, 2, and 5 (*Figure 4—figure supplement 2Ai-iv*). F480⁺MHCIIʰⁱ IMs expressed iRFP682/CCR5, but Clover/CCR1⁺ and mRuby2/CCR2⁺ cells were less abundant (*Figure 4—figure supplement 2Bi-iv*). This pattern is similar to that of MHCIIᴸᵒ IMs, suggesting that F480⁺MHCIIʰⁱ IMs also downregulate CCR2 as they differentiate from F480ᴸᵒMHCIIʰⁱ IMs. In line with this, co-expression of Clover/CCR1 or iRFP682/CCR5 with mRuby2/CCR2 was more apparent in F480ᴸᵒMHCIIʰⁱ than in F480⁺MHCIIʰⁱ IMs (*Figure 4—figure supplement 2Ci-iv*).

To further confirm this pattern, we generated GM-CSF-derived macrophages and DCs from BM of iCCR-REP mice. As shown in *Figure 4—figure supplement 2*, and in line with previous findings, freshly extracted BM Ly6Cʰⁱ monocytes expressed mRuby2/CCR2 almost exclusively, with only a small fraction co-expressing Clover/CCR1 (*Figure 4—figure supplement 2D*). After 2 days in culture with GM-CSF, Ly6Cʰⁱ monocytes retain expression of mRuby2/CCR2 but induce Clover/CCR1 and iRFP682/CCR5, with approximately 40% of the cells co-expressing all three reporters (*Figure 4—figure supplement 2D*). At this time point, CD11c⁺ precursors are already detectable in cultures. These precursors express Clover/CCR1 and iRFP682/CCR5 to a higher extent than observed in Ly6Cʰⁱ monocytes, with 65% of the population co-expressing reporters for CCRs 1, 2, and 5 (*Figure 4—figure supplement 2D*). By day 9 in culture, fully mature CD11c⁺/MHCII⁺⁺ monocyte-derived dendritic cells (moDCs) express mainly Clover/CCR1 and iRFP682/CCR5, which are now detected independently from mRuby2/CCR2 in over 50% of the cells (*Figure 4—figure supplement 2D*). Finally, we tested the monocyte downregulation of mRuby2/CCR2 and macrophage upregulation of Clover/CCR1 and iRFP682/CCR5 in vivo. To this end (*Figure 4—figure supplement 3A and B*), we adoptively transferred mRuby2/CCR2+ve monocytes (which are -ve for iRFP682/CCR5, and only a small fraction expresses Clover/CCR1) from the reporter mice into WT air-pouches. We then reclaimed cells 3 days later and examined macrophages for reporter expression by flow cytometry. As shown in *Figure 4—figure supplement 3*, and in contrast to the adoptively transferred precursor monocytes, the progeny macrophages now express substantial levels of Clover/CCR1 and iRFP682/CCR5 reporters. This experiment further reinforces the notion that CCR2+ve monocytes progressively express CCR1 and CCR5 as they differentiate to macrophages.

At rest, lung eosinophils only expressed mTagBFP2/CCR3 (*Figure 4—figure supplement 2Di-ii*), whereas alveolar macrophages did not express any of the iCCR reporters (data not shown). We did not detect expression of any of the iCCR reporters in lung or kidney neutrophils (data not shown).

Thus, these data demonstrate that, whilst classical monocytes (in contrast to nonclassical monocytes) predominantly express CCR2, they downregulate this receptor and upregulate CCR1 and CCR5 as they differentiate. In keeping with the observations from blood and BM, eosinophils within tissues are positive only for CCR3.

## iCCR expressing cells can be directly visualised in tissues

The above analyses focused on flow cytometric evaluation of iCCR-REP expression. However, we next determined whether these mice are also appropriate for direct visualisation and localisation of

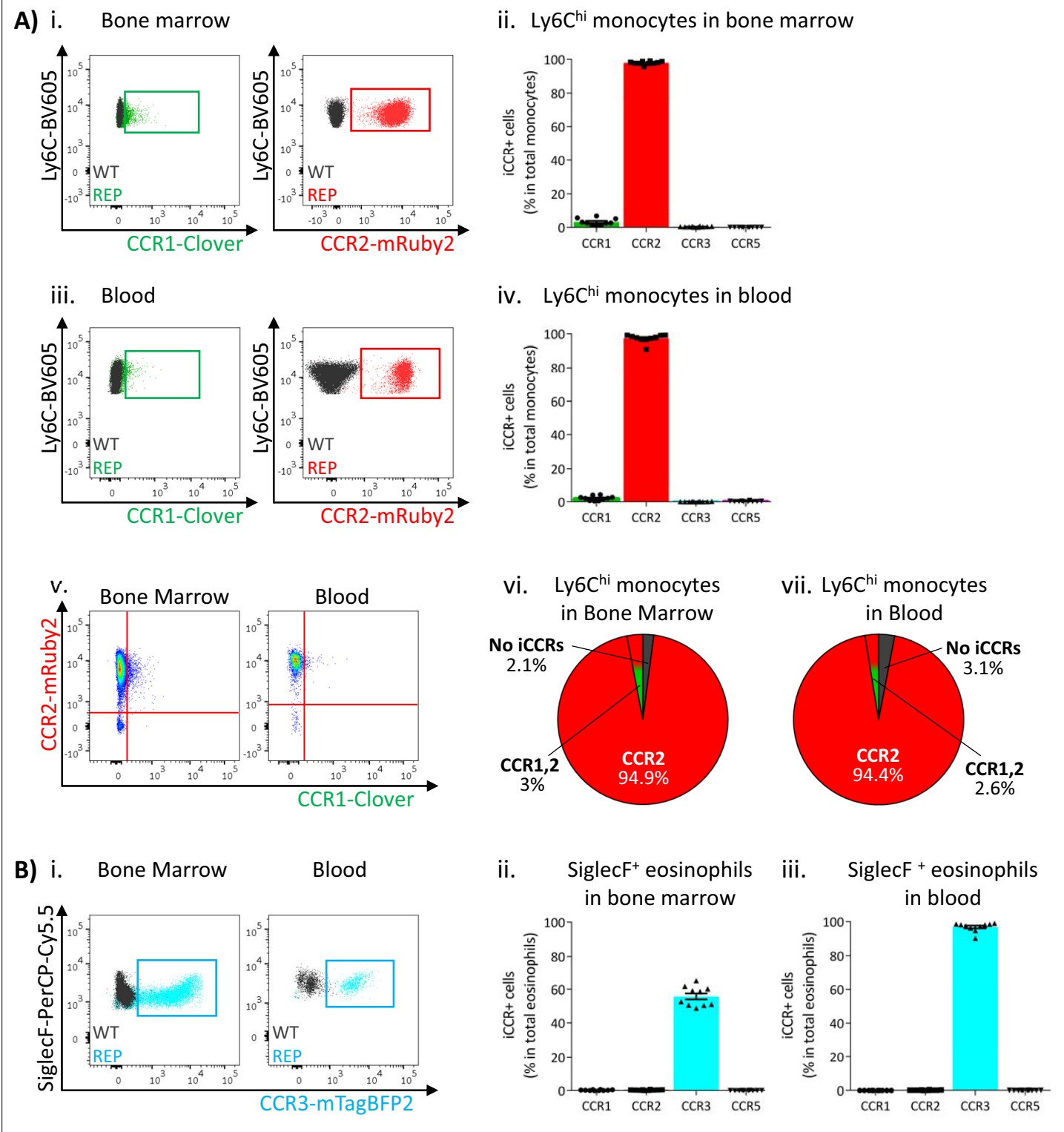

**Figure 3.** Inflammatory CC chemokine receptor (iCCR) expression in bone marrow and blood leukocytes at rest. (**A**) Flow cytometric analysis of Clover/CCR1 and mRuby2/CCR2 expression in (i) bone marrow and (iii) circulating Ly6C^hi monocytes. Quantification of the percentage of Ly6C^hi monocytes expressing the fluorescent reporters in (ii) bone marrow and (iv) blood. (v) Flow cytometric analysis of Clover/CCR1 and mRuby2/CCR2 co-expression in bone marrow and circulating Ly6C^hi monocytes. Distribution of Clover and mRuby2 in (vi) bone marrow and (vii) circulating Ly6C^hi monocytes. (**B**) (i) Flow cytometric analysis of mTagBFP2/CCR3 expression in bone marrow and circulating SiglecF^+ eosinophils. Quantification of the percentage of SiglecF^+ eosinophils expressing the iCCR reporters in (ii) bone marrow and (iii) blood. Data on A–B are compiled from at least three separate experiments. Data

*Figure 3 continued on next page*

*Figure 3 continued*

on Aii, Aiv, Bii, and Biii are shown as mean ± SEM (N=10). Each data point represents a measurement from a single mouse. Blots in Ai, Aiii, and Bi are combinatorial blots showing reporter expression in iCCR-REP and wild type (WT) (control for background autofluorescence) mice.

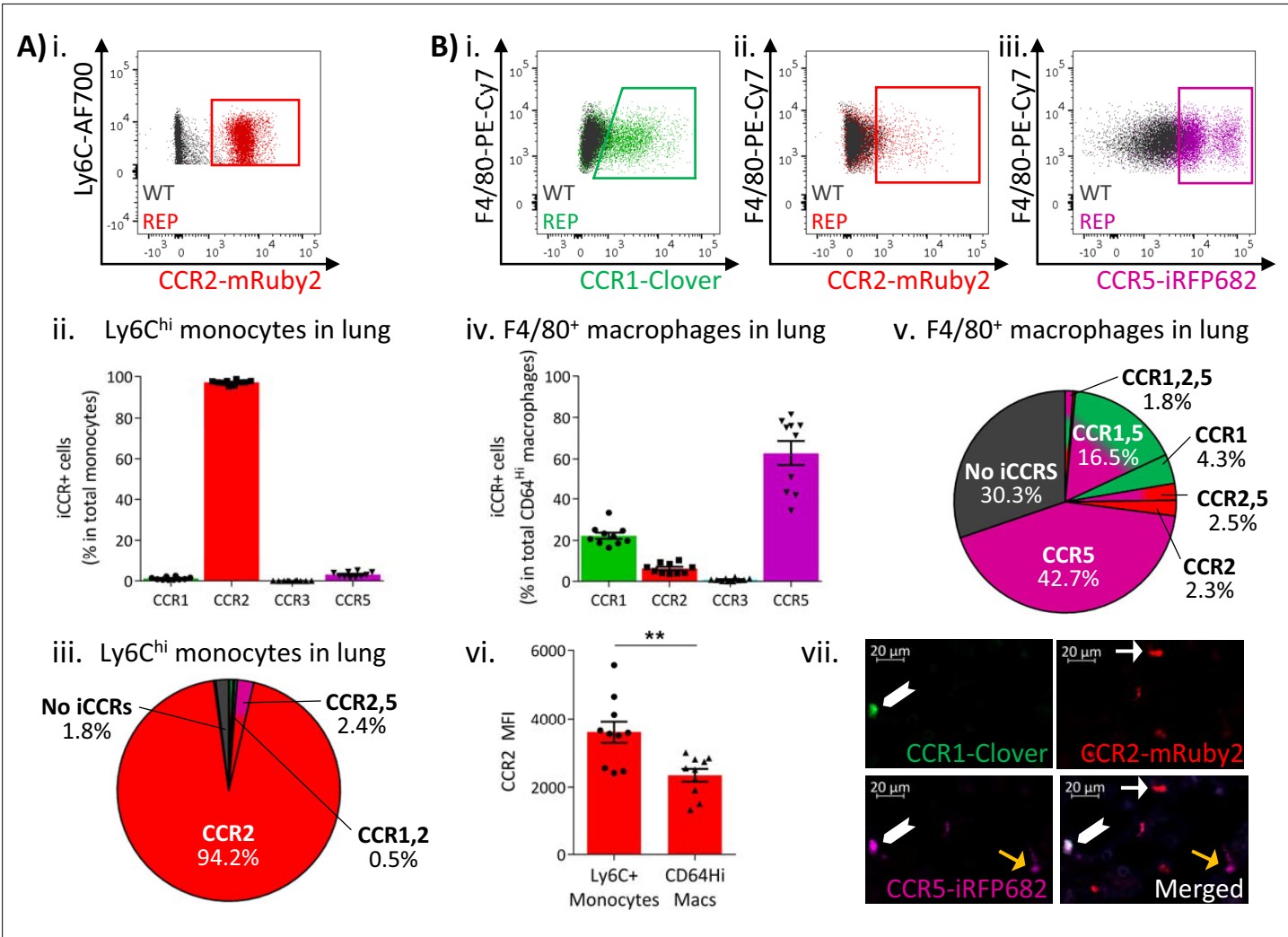

**Figure 4.** Inflammatory CC chemokine receptor (iCCR) expression in resting lung. (**A**) (i) Flow cytometric analysis of mRuby2/CCR2 expression in Ly6C^hi monocytes. (ii) Quantification of the percentage of Ly6C^hi monocytes expressing the iCCR reporters. (iii) Distribution of the iCCR reporters in Ly6C^hi monocytes. (**B**) Flow cytometric analysis of (i) Clover/CCR1, (ii) mRuby2/CCR2, and (iii) iRFP682/CCR5 expression in F4/80+ macrophages. (iv) Quantification of the percentage of F4/80+ macrophages expressing the iCCR reporters. (v) Distribution of the iCCR reporters in F4/80+ macrophages. (vi) mRuby2/CCR2 mean fluorescence intensity from Ly6C^hi monocytes and F4/80+ macrophages. (vii) Lung leukocytes expressing CCR2 exclusively (white arrow), CCR1 and CCR5 (chevron) or CCR5 exclusively (yellow arrow). Data in A–B are compiled from at least two separate experiments. Data on Aii, Biv, Bvi are shown as mean ± SEM (N=10). Each data point represents a measurement from a single mouse. Blots in Ai, Bi, Bii, and Biii are combinatorial blots showing reporter expression in iCCR-REP and wild type (WT) (control for background autofluorescence) mice. Data on Bvi were analysed using unpaired t-test. \*\*p<0.01. See also *Figure 4—figure supplement 2* and , *Figure 4—figure supplement 4*.

The online version of this article includes the following figure supplement(s) for figure 4:

**Figure supplement 1.** Reporter co-expression in resting tissues.

**Figure supplement 2.** Inflammatory CC chemokine receptor (iCCR) expression in resting lung.

**Figure supplement 3.** Inflammatory CC chemokine receptor (iCCR) expression in monocytes and macrophages during differentiation.

**Figure supplement 4.** Inflammatory CC chemokine receptor (iCCR) reporters are readily visualised in tissues.

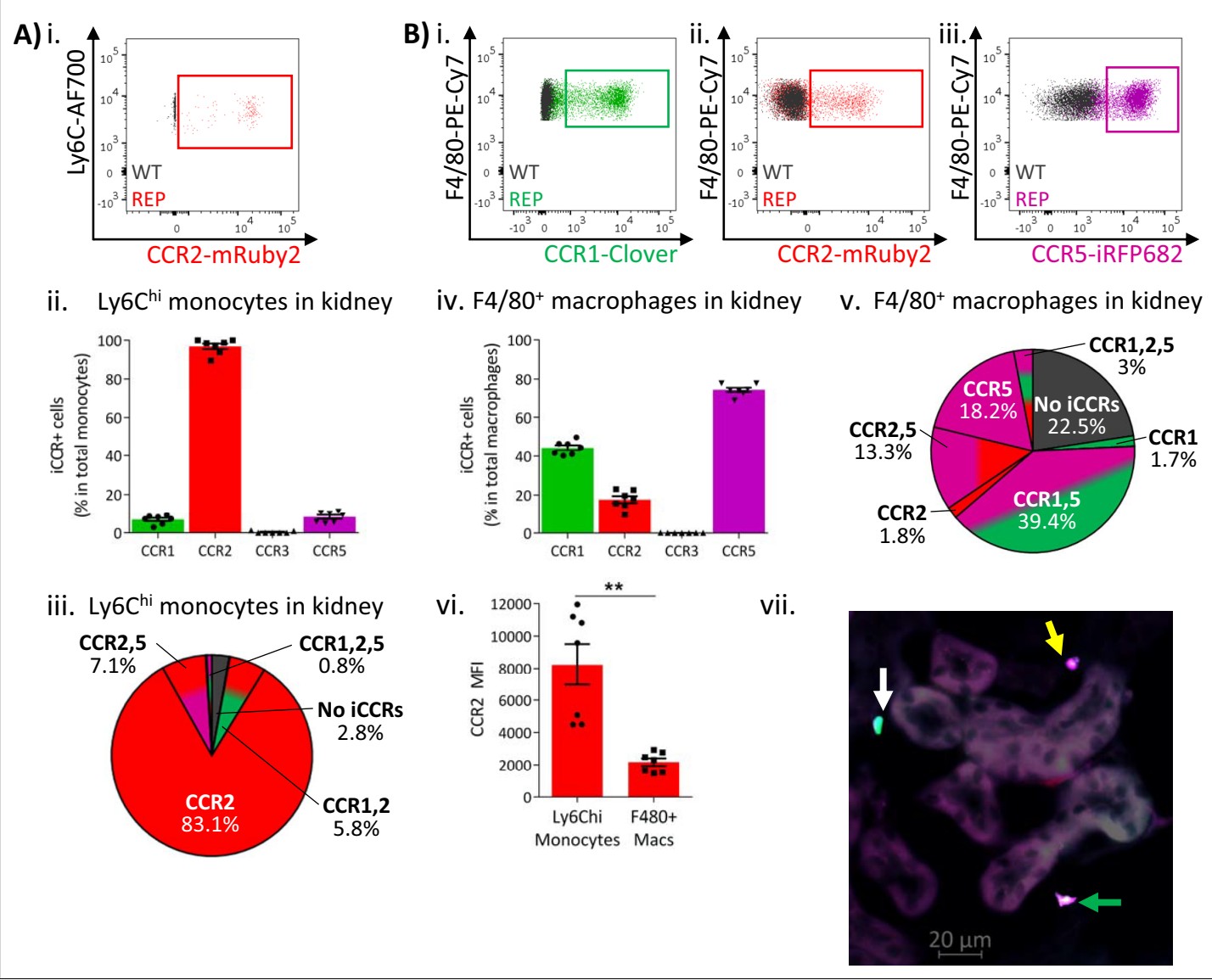

**Figure 5.** Inflammatory CC chemokine receptor (iCCR) expression in resting kidney. (**A**) (i) Flow cytometric analysis of mRuby2/CCR2 expression in Ly6C^hi monocytes. (ii) Quantification of the percentage of Ly6C^hi monocytes expressing the iCCR reporters. (iii) Distribution of the iCCR reporters in Ly6C^hi monocytes. (**B**) Flow cytometric analysis of (**i**) Clover/CCR1, (ii) mRuby2/CCR2, and (iii) iRFP682/CCR5 expression on F4/80^+ macrophages. (iv) Quantification of the percentage of F4/80^+ macrophages expressing the iCCR reporters. (v) Distribution of the iCCR reporters in F4/80^+ macrophages. (vi) mRuby2/CCR2 mean fluorescence intensity from Ly6C^hi monocytes and F4/80^+ macrophages. (vii) Kidney leukocytes expressing CCR1 exclusively (white arrow), CCR5 exclusively (yellow arrow) or CCR1 and CCR5 (green arrow). Data in A–B are compiled from at least two separate experiments. Data on Cii, Civ, and Cvi are shown as mean ± SEM (N=7). Each data point represents a measurement from a single mouse. Blots in Ai, Bi, Bii, and Biii are combinatorial blots showing reporter expression in iCCR-REP and wild type (WT) (control for background autofluorescence) mice. Data on Bvi were analysed using unpaired t-test with Welch's correction. **p<0.01. See also *Figure 5—figure supplement 1* and *Figure 5—figure supplement 2*.

The online version of this article includes the following figure supplement(s) for figure 5:

**Figure supplement 1.** Reporter co-expression in resting tissues.

**Figure supplement 2.** Inflammatory CC chemokine receptor (iCCR) reporters visualised in resting kidney.

iCCR-reporter expressing cells within tissues. As shown in *Figure 4—figure supplement 4*, fluorescence imaging of a section from resting spleen revealed easily identifiable cells expressing each of the four reporters. In addition, combinatorial iCCR reporter expression was detected on individual cells as shown in *Figure 4Bvii*. Here, we highlight lung cells expressing only mRuby2/CCR2 as well as cells that are co-expressing Clover/CCR1 with iRFP682/CCR5. Finally, the imaging approaches are applicable to

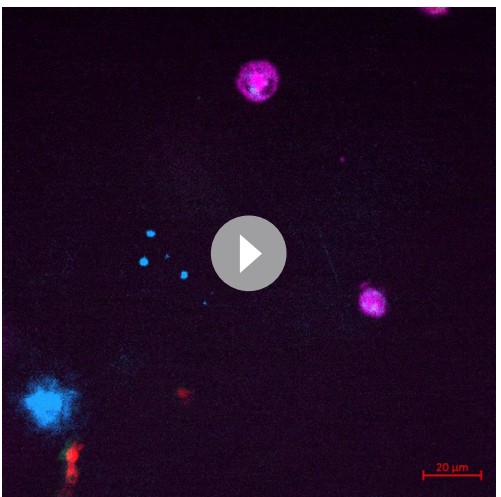

**Video 1.** Real-time in vivo imaging of iCCR reporters in resting mammary gland.

https://elifesciences.org/articles/72418/figures#video1

numerous tissues and, as shown in *Figure 5Bvii*, iCCR-REP mice can be used to identify leukocyte populations expressing individual and combinatorial patterns of iCCR expression in the resting kidney (note that this image has been enhanced, using SPARCA software, to remove excessive autofluorescence from the renal tubules. The original image is shown in *Figure 5—figure supplement 2*).

In addition to static imaging, the iCCR-REP mice can be used for real-time imaging of reporter expression in living tissues. As shown in the movie in *Video 1*, real-time in vivo imaging of the mouse mammary gland demonstrates the ability of the mice to be used to examine the dynamics of leukocyte movement within a tissue.

## iCCR expression in BM and circulation under inflammatory conditions

To determine myeloid cell iCCR reporter expression under inflamed conditions, we first used the air-pouch model (*Figure 6A*). iCCR reporter expression in BM and blood was assessed 48 hr after the induction of inflammation (gating strategies: *Figure 1—figure supplement 1B*). Similar to resting conditions, the majority of Ly6C$^{hi}$ monocytes expressed mRuby2/CCR2 and a fraction expressed Clover/CCR1 (*Figure 6Bi–ii* and 6Biv–v). However, the fraction of Ly6C$^{hi}$ monocytes expressing Clover/CCR1 was higher than in resting mice. Approximately 20% of Ly6C$^{hi}$ monocytes expressed Clover/CCR1 in the BM of inflamed mice, representing a sixfold increase compared to resting conditions (*Figure 6Biii*). Similarly, in blood, approximately 11% of Ly6C$^{hi}$ monocytes expressed Clover/CCR1, representing a fivefold increase compared with resting conditions (*Figure 6Bvi*). Interestingly, in this inflamed context, we also detected, for the first time, expression of Clover/CCR1 independently of mRuby2/CCR2 on monocytes, although in most cases both iCCR reporters are co-expressed (*Figure 6Ci–iii*). BM and blood eosinophils showed similar iCCR reporter expression to that observed under resting conditions. Only mTagBFP2/CCR3 was detected, with approximately 50% of eosinophils expressing it in the BM and 85% in the circulation (*Figure 6D*).

These results suggested that sustained inflammation leads to enhanced CCR1 expression on BM and circulating monocytes. We therefore used a different model to test this hypothesis. We have previously seen (data not shown) that Clover/CCR1 expression can be upregulated following in vitro interferon gamma (IFNγ) treatment of cells (data not shown). We therefore implanted IFNγ-loaded subcutaneous osmotic pumps (or PBS control) in iCCR-REP mice providing continuous IFNγ release into the circulation (*Figure 7A*). After 7 days, BM and blood monocytes were analysed to determine expression of Clover/CCR1. As shown in *Figure 7Bi–ii*, we observed a sixfold increase in the number of Clover/CCR1 expressing monocytes in BM after IFNγ treatment. Notably, this was not reflected in a similar increase in Clover/CCR1 MFI (*Figure 7Biii*). Similarly, in peripheral blood, a trend towards increased Clover/CCR1$^+$ monocytes was observed (*Figure 7Biv and v*), although this was not statistically significant and again no difference in MFI was seen (*Figure 7Bvi*). In all cases, Clover/CCR1 was co-expressed with mRuby2/CCR2 in this model (*Figure 7C*).

Together, these results confirm that inflammation is associated with the accumulation of Clover/CCR1$^+$ monocytes in BM and circulating monocytes.

## iCCR expression in inflamed tissues: the air-pouch model

We next analysed myeloid cell iCCR reporter expression in the inflamed air-pouch. We first examined the membrane that surrounds the inflamed cavity (gating strategies: *Figure 1—figure supplement 1C*). Recently infiltrated Ly6C$^{hi}$ monocytes and CD11b$^+$F480$^+$ macrophages retain expression of mRuby2/CCR2 (*Figure 8A* and *Figure 8—figure supplement 1*), indicative of the rapid turnover of this population under inflammatory conditions. Approximately 20% of Ly6C$^{hi}$ monocytes expressed

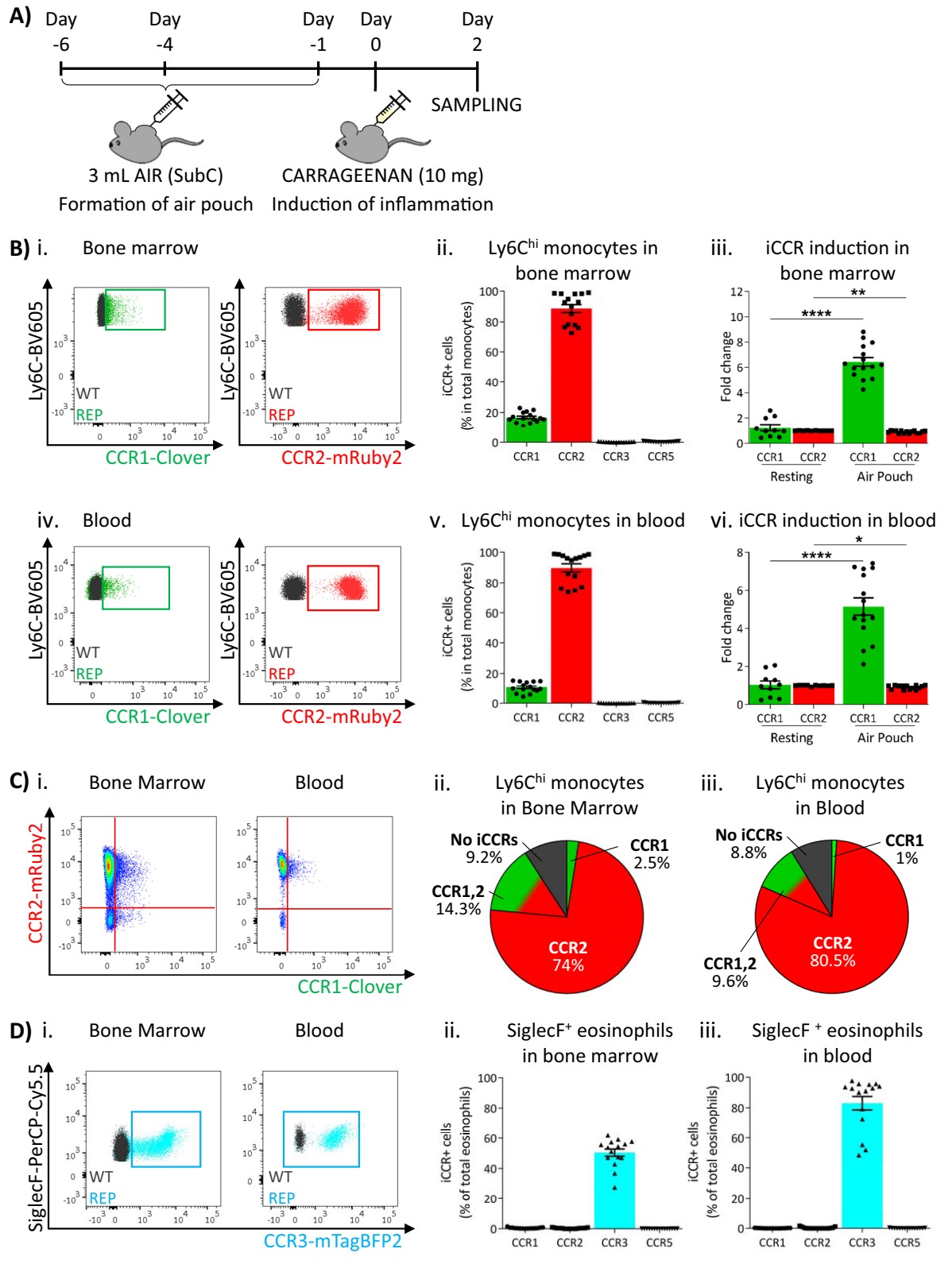

**Figure 6.** Inflammatory CC chemokine receptor (iCCR) expression in acutely inflamed bone marrow and blood leukocytes. (**A**) Schematic of the procedure used to induce acute inflammation using the air-pouch model. (**B**) Flow cytometric analysis of Clover/CCR1 and mRuby2/CCR2 expression in (i) bone marrow and (iv) circulating Ly6C$^{hi}$ monocytes. Quantification of the percentage of (ii) bone marrow and (v) circulating Ly6C$^{hi}$ monocytes expressing the iCCR reporters. Quantification of the fold change increase in CCR1 and CCR2 expression by (iii) bone marrow and (vi) circulating Ly6C$^{hi}$

*Figure 6 continued on next page*

Figure 6 continued

monocytes after induction of inflammation. (**C**) (i) Flow cytometric analysis and distribution of Clover/CCR1 and mRuby2/CCR2 in (ii) bone marrow and (iii) circulating Ly6C$^{hi}$ monocytes in the air-pouch model. (**D**) (i) Flow cytometric analysis of mTagBFP2/CCR3 expression in bone marrow and circulating SiglecF$^+$ eosinophils. Quantification of the percentage of SiglecF$^+$ eosinophils expressing the iCCR reporters in (ii) bone marrow and (iii) blood. Data on Bii, Biii, Bv, Bvi, Dii, and Diii are shown as mean ± SEM (N=10 for resting mice or N=15 for carrageenan-treated mice) and are compiled from at least three separate experiments. Each data point represents a measurement from a single mouse. Blots in Bi, Biv, and Di are combinatorial blots showing reporter expression in iCCR-REP and wild type (WT) (control for background autofluorescence) mice. Normally distributed data on Biii and Bvi were analysed using unpaired t-test with or without Welch's correction, according to their standard deviations. Not normally distributed data were analysed using Mann-Whitney or Kolmogorov-Smirnov, according to the standard deviations. *p<0.05; **p<0.01; ****p<0.0001.

Clover/CCR1 (*Figure 8Aii*), whereas expression of this receptor was much lower (*Figures 4 and 5*) in intra-tissue monocytes under resting conditions. Note that for mRuby2/CCR2 and mTagBFP2/CCR3, a bimodal level of fluorescent reporter expression is seen in these figures. This is an artefact of the fixation required for air-pouch membrane analysis and is not seen with unfixed cells (*Figure 8—figure supplement 1*). The lower intensity MFI population is the artefactual one. Notably, this figure also demonstrates that iRFP682/CCR5 is also affected by fixation in a way that is not seen on analysis of unfixed cells. Clover/CCR1 expression was further increased as Ly6C$^{hi}$ monocytes differentiated into CD11b$^+$F480$^+$ macrophages, with approximately 40% of this population now expressing the receptor (*Figure 8Aiii* and iv). These data, together with the induction of Clover/CCR1 in inflamed BM and blood monocytes, suggest a role for this receptor in monocyte recruitment and macrophage migration in this inflammation model. In contrast, iRFP682/CCR5 expression was confined to a small fraction of the monocyte and macrophage populations (*Figure 8A*), whereas its expression was abundant on resting tissue macrophages (*Figures 4 and 5*). This suggests a less significant contribution of CCR5 to monocyte and macrophage motility in the air-pouch model.

As shown in *Figure 8B*, eosinophils in the air-pouch retain exclusive expression of mTagBFP2/CCR3, confirming the importance of this receptor for eosinophil recruitment to inflamed sites. We did not detect expression of any iCCR reporter in neutrophils (data not shown).

Analysis of NK cells and lymphoid subsets (*Figure 8—figure supplement 2*), in the air-pouch, indicated expression of both mRuby2/CCR2 and iRFP682/CCR5 by CD11b$^+$ and CD11b$^-$ NK cells. In addition, mRuby2/CCR2, and low-level iRFP682/CCR5, expression was detected on NKT cells and mRuby2/CCR2 was the only one of the reporters to be detected on either αβ or γδ T cells in this model. In a separate study, looking at the tumour microenvironment, we have detected iRFP682/CCR5 on T cells (data not shown). We have not, so far, detected Clover/CCR1 on any T cell populations.

## iCCR expression in inflamed tissues: the intranasal LPS model

To determine if the pattern of iCCR reporter expression detected in the air-pouch model was also seen in a more relevant inflammatory model, we next used an LPS model of lung inflammation. Here, LPS (or vehicle) was administered intranasally to iCCR-REP mice or WT littermates (*Figure 9Ai*). Forty-eight hours later, inflamed lungs were dissected and myeloid cell content examined (gating strategies: *Figure 1—figure supplement 2A*). Fluorescence imaging of lung sections from PBS and LPS-treated iCCR-REP mice (*Figure 9—figure supplement 1*) demonstrates sparse reporter expression in control lungs but widespread expression of the reporters, especially Clover/CCR1, mRuby2/CCR2, and iRFP682/CCR5. In keeping with our previous results showing lack of eosinophil recruitment in response to acute LPS, very little expression of the CCR3 reporter mTagBFP2 was seen. Infiltration of Ly6C$^{hi}$ monocytes and CD11b$^+$F480$^+$ IMs into inflamed lung was confirmed by flow cytometry (*Figure 9Aii*). Consistent with the findings from the air-pouch model, Ly6C$^{hi}$ monocytes and CD11b$^+$F480$^+$ IMs both retain expression of mRuby2/CCR2, indicative of their recent infiltration into the inflamed lung (*Figure 9Aiii-iv*). Similarly, 23% of the lung monocyte population expressed Clover/CCR1 after LPS treatment, compared to only 3.5% of monocytes in vehicle-treated mice (*Figure 9Aiii*). This level of expression is maintained in mature CD11b$^+$F480$^+$ IMs from LPS-treated lungs (*Figure 9Aiv, B and C* and *Figure 9—figure supplement 2*), confirming the rapid induction of CCR1 in infiltrated Ly6C$^{hi}$ monocytes. Finally, also consistent with observations in the air-pouch model, iRFP682/CCR5 was detected in only 7.5% of Ly6C$^{hi}$ monocytes and 12% of CD11b$^+$F480$^+$ IMs after LPS administration, whereas 65% of CD11b$^+$F480$^+$ IMs expressed it in the vehicle-treated lungs (*Figure 9Aiii-iv, B and C*). These data suggest that resident CCR5$^+$ IMs are rapidly replaced with CCR1$^+$ IMs after induction

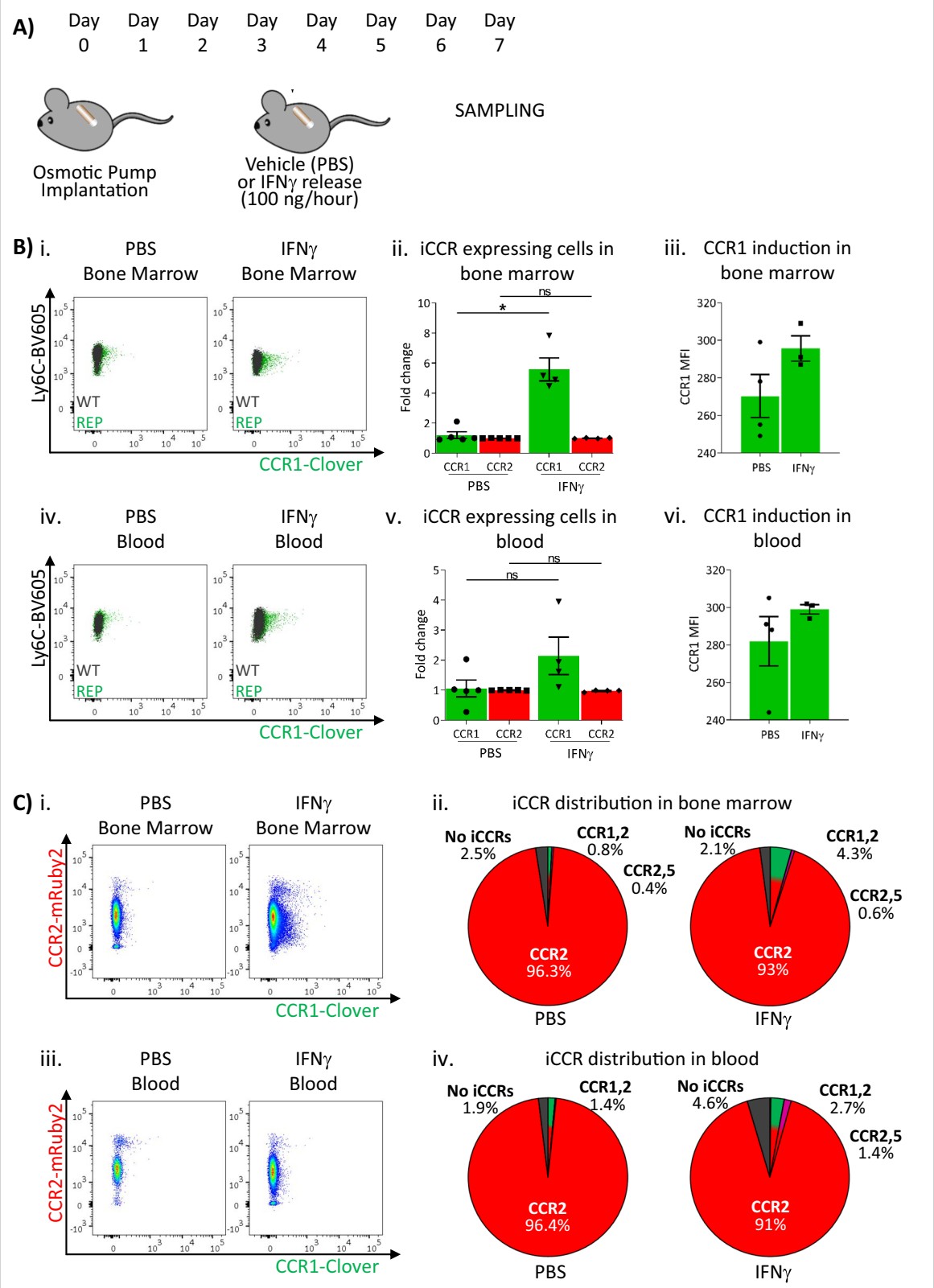

**Figure 7.** Inflammatory CC chemokine receptor (iCCR) expression in bone marrow and blood under sustained inflammation. (**A**) Schematic of the procedure used to induce sustained inflammation using interferon gamma (IFNγ) release from osmotic pumps. (**B**) Flow cytometric analysis of Clover/ CCR1 expression in (i) bone marrow and (iv) circulating Ly6C$^{hi}$ monocytes. Quantification of the fold change increase in CCR1$^+$ and CCR2$^+$ cells in (ii) bone marrow and (v) circulating Ly6C$^{hi}$ monocytes after the induction of sustained inflammation. Mean fluorescence intensity of Clover/CCR1 expression

*Figure 7 continued on next page*

*Figure 7 continued*

in (iii) bone marrow and (vi) circulating CCR1⁺ Ly6 Cʰⁱ monocytes. (**C**) Flow cytometric analysis and distribution of Clover/CCR1 and mRuby2/CCR2 in (i and ii) bone marrow and (iii and iv) circulating Ly6Cʰⁱ monocytes. Data on B–C are compiled from at least two separate experiments. Data on Bii, Biii, Bv, and Bvi are shown as mean ± SEM (N=3-5). Each data point represents a measurement from a single mouse. Plots in Bi and Biv are combinatorial plots showing reporter expression in iCCR REP and wild type (WT) (control for background autofluorescence) mice. Normally distributed data were analysed using unpaired t-test with or without Welch's correction, according to their standard deviations. Not normally distributed data were analysed using Mann-Whitney or Kolmogorov-Smirnov, according to the standard deviations. *p<0.05; ns, not significant.

of inflammation and indicate a less significant contribution of CCR5 to monocyte and macrophage recruitment and migration in the early stages of the inflammatory response to LPS.

We also evaluated iCCR expression on SiglecF⁺ eosinophils and SiglecF⁺F480⁺ alveolar macrophages after LPS inoculation. As discussed above, eosinophils are not recruited into the lung in acute models of LPS exposure. However, all detected eosinophils in control and LPS inflamed lungs showed exclusive expression of mTagBFP2/CCR3 (*Figure 9Di*). Alveolar macrophages do not express any iCCR reporter after vehicle treatment (resting conditions). However, after LPS administration, 50% of the population shows expression of Clover/CCR1 (*Figure 9Dii*), suggesting that this receptor is also important for their intra-tissue function under inflammatory conditions. We did not detect expression of iCCR reporters in neutrophils from inflamed lungs (data not shown).

## Discussion

The iCCRs are responsible for the mobilisation, recruitment, and intra-tissue dynamics of all non-neutrophilic myeloid cell subsets as well as some lymphoid subsets. Analysis of their in vivo expression dynamics has been hampered by the difficulties associated with generating combinatorial reporter mice using individual reporter strains (*Hirai et al., 2014*; *Luckow et al., 2009*; *Saederup et al., 2010*), due to their genomic proximity and the incompatibility of the reporters used in these strains. We now report the generation, and analysis, of transgenic mice expressing spectrally distinct fluorescent reporters for each of the four iCCRs. To our knowledge, this is the first mouse model allowing simultaneous, and combinatorial, analysis of four different fluorescent reporters for the study of specific protein expression, and provides a template for the generation of similar reporter mice covering other functionally linked genomic loci.

The iCCR-REP mice are viable, display normal leukocyte numbers, and thus there appear to be no deleterious effects of the BAC transgene. With any reporter mouse model there is the concern that expression of the reporter molecule may not accurately reflect the expression of the actual protein being assessed. This can be a consequence of posttranscriptional events or a disconnect between the half-lives of the reporters and of the molecules being studied. However, we have gone to some lengths to demonstrate close correlation between reporter expression and chemokine receptor protein expression where possible. Specifically, we demonstrate that reporter expression accurately reflects surface iCCR presentation and that expression changes under different inflammatory conditions and as monocytes differentiate. Previously, antibodies have been widely used to study iCCR expression. However, our results (using appropriate iCCR⁻/⁻ mice as controls) demonstrate that these antibodies show background non-specific staining due to the high degree of homology between different iCCRs. Lack of such controls has frequently been a problem in previous antibody-based analyses. Also in our hands, and as described by others (*Hirai et al., 2014*), commercially available antibodies for CCR1 do not bind efficiently to the receptor. In addition, when looking at cells isolated from enzyme digested tissues, the external portions of chemokine receptors are frequently cleaved by the enzymes, preventing antibody-dependent detection. In contrast, our iCCR-REP mice display highly specific expression of the reporters, representing a powerful tool for the flow cytometric, and imaging-based, analysis of in vivo iCCR expression.

Importantly, the analysis of the iCCR-REP mice has allowed us, for the first time, to unequivocally establish iCCR expression patterns on myeloid cells at rest and during inflammatory responses. In contrast to previous reports of multiple receptor expression by individual leukocyte subtypes (*Haringman et al., 2006*; *Tacke et al., 2007*; *Weber et al., 2000*), our data indicate that iCCR expression on individual cell subsets in resting mice is selective. Thus, most Ly6Cʰⁱ monocytes in BM and blood exclusively express CCR2, with only a minor fraction of these cells co-expressing CCR1 and

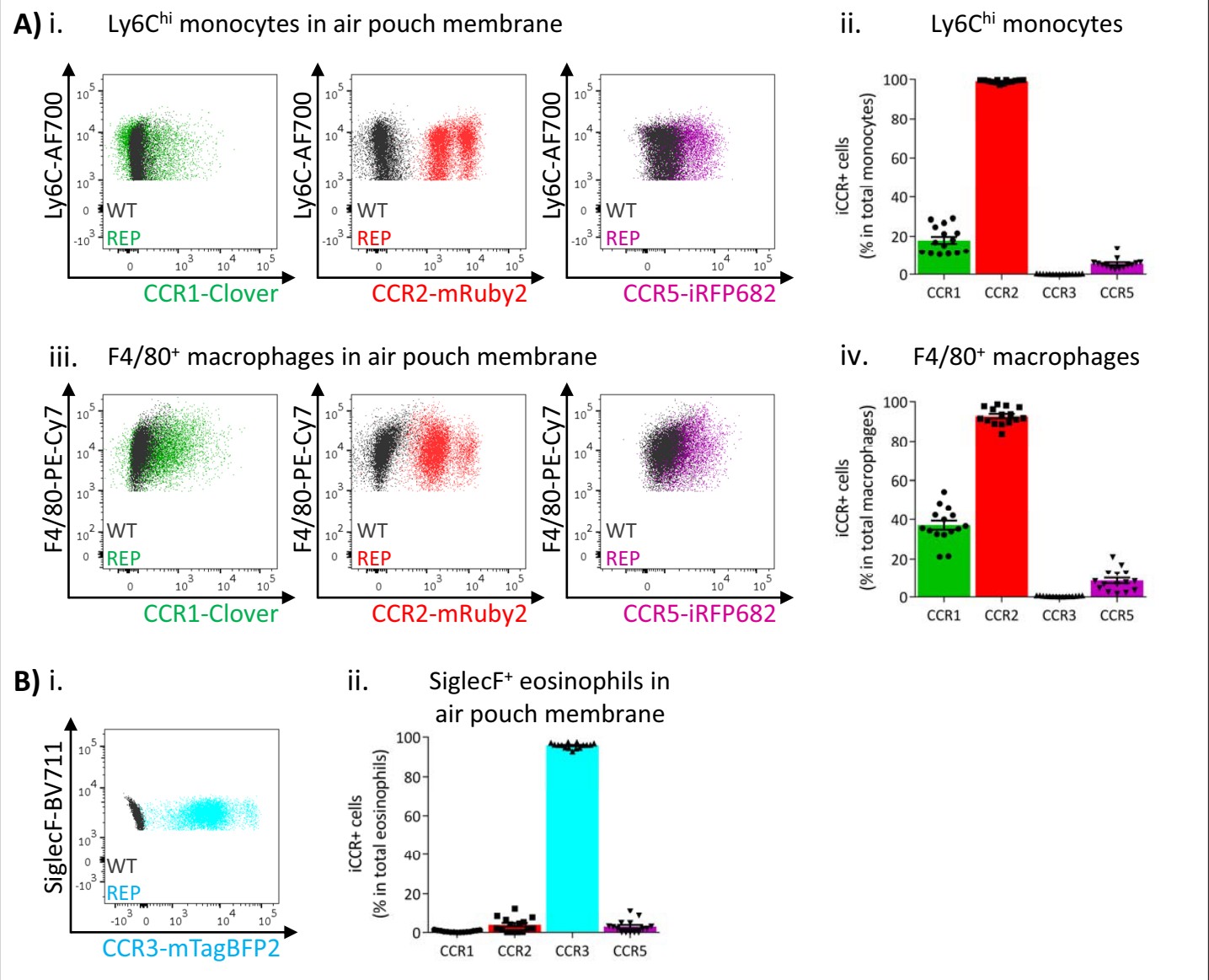

**Figure 8.** Inflammatory CC chemokine receptor (iCCR) expression in inflamed tissues: the air-pouch model. (**A**) Flow cytometric analysis of Clover/CCR1, mRuby2/CCR2, and iRFP682/CCR5 expression in (**i**) Ly6C$^{hi}$ monocytes and (**iii**) F4/80$^+$ macrophages isolated from the inflamed air-pouch. Quantification of iCCR reporter expression in (**ii**) Ly6C$^{hi}$ monocytes and (**iv**) F4/80$^+$ macrophages. (**B**) (**i**) Flow cytometric analysis of mTagBFP2/CCR3 expression in SiglecF$^+$ eosinophils isolated from the inflamed air-pouch. (ii) Quantification of iCCR reporter expression in SiglecF$^+$ eosinophils. Data on Aii, Aiv, and Bii are shown as mean ± SEM (N=15) and are compiled from at least three separate experiments. Each data point represents a measurement from a single mouse. Blots in Ai, Aiii, and Bi are combinatorial blots showing reporter expression in iCCR-REP and wild type (WT) (control for background autofluorescence) mice.

The online version of this article includes the following figure supplement(s) for figure 8:

**Figure supplement 1.** Reporter co-expression in inflamed air-pouch.

**Figure supplement 2.** Inflammatory CC chemokine receptor (iCCR) expression in inflamed air-pouch: lymphoid cells.

CCR2. Eosinophils only express CCR3. Tissue-resident macrophages downregulate CCR2 and express CCR1 and high levels of CCR5 either alone or in combination. This suggests a hierarchical relationship in which monocytes use CCR2 to egress from BM and infiltrate resting tissues but upregulate CCR1 and CCR5 upon differentiation to macrophages or moDCs. This model was further confirmed by our in vitro studies using GM-CSF BM-derived moDCs. While Ly6C$^{hi}$ monocytes freshly isolated from BM expressed almost exclusively CCR2, culture in GM-CSF containing media induced gradual

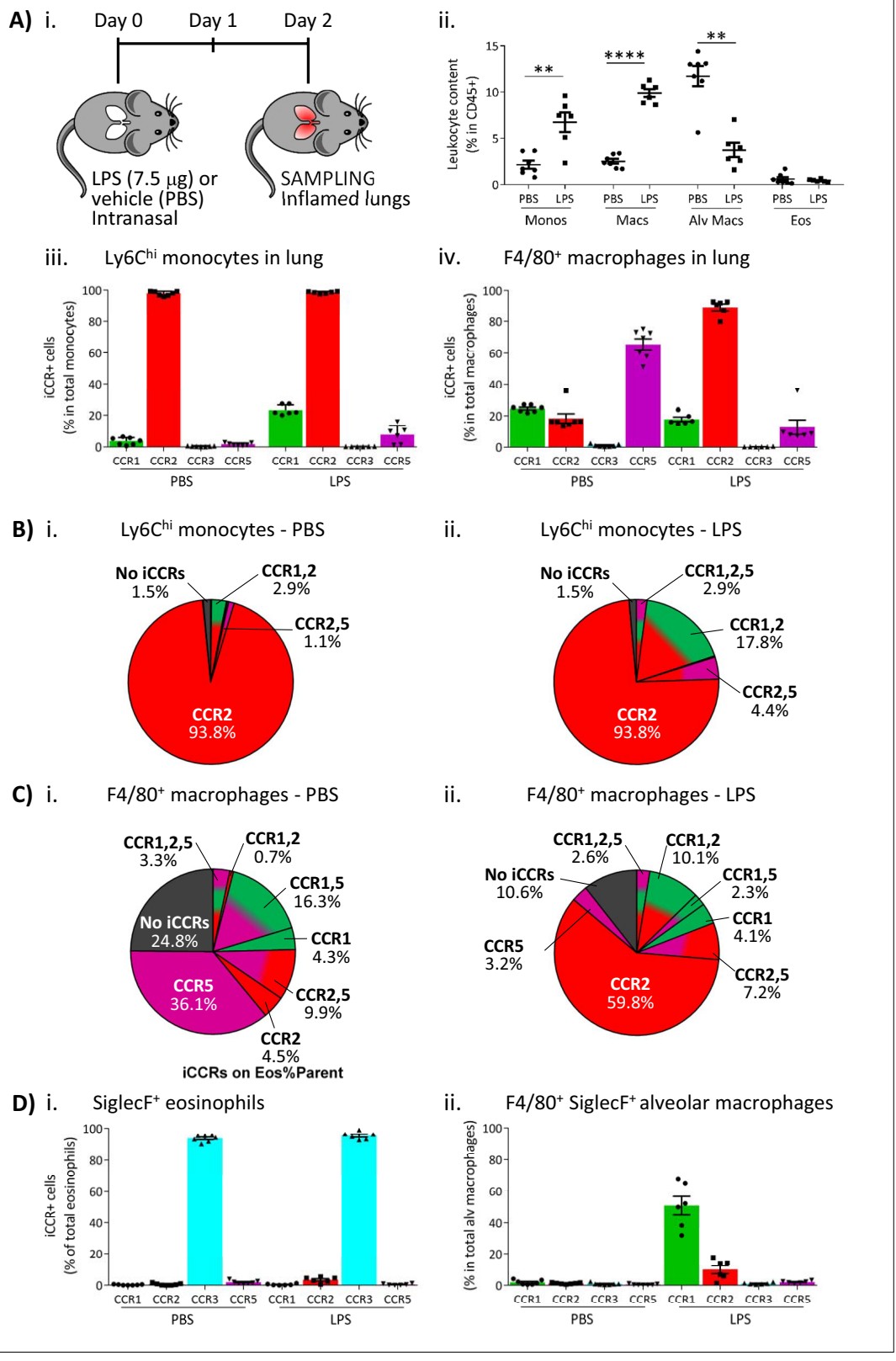

**Figure 9.** Inflammatory CC chemokine receptor (iCCR) expression in inflamed tissues: the intranasal lipopolysaccharide (LPS) model. (**A**) (i) Schematic of the procedure used to induce acute lung inflammation using intranasal administration of LPS. (ii) Quantification of monocyte, macrophage, alveolar macrophage, and eosinophil levels in inflamed lungs. Quantification of iCCR reporter expression in (iii) Ly6C$^{hi}$ monocytes and (iv) F4/80$^{+}$

*Figure 9 continued on next page*

*Figure 9 continued*

macrophages isolated from lungs of vehicle (PBS) and LPS-treated mice. (**B**) Distribution of Clover/CCR1, mRuby2/CCR2, and iRFP682/CCR5 on Ly6C$^{hi}$ monocytes isolated from lungs of (i) vehicle (PBS) and (ii) LPS-treated mice. (**C**) Distribution of Clover/CCR1, mRuby2/CCR2, and iRFP682/CCR5 on F4/80$^+$ macrophages isolated from lungs of (i) vehicle (PBS) and (ii) LPS-treated mice. (**D**) Quantification of iCCR reporter expression on (**i**) SiglecF$^+$ eosinophils and (**ii**) SiglecF$^+$ F4/80$^+$ alveolar macrophages isolated from lungs of vehicle and LPS-treated mice. Data in A–D are compiled from at least two separate experiments. Data on Aii, Aiii, Aiv, Di, and Dii are shown as mean ± SEM (N=7 for vehicle-treated mice or N=6 for LPS-treated mice). Each data point represents a measurement from a single mouse. Data on Aii were analysed using unpaired t-test, with the exception of alveolar macrophage, which was analysed using Mann-Whitney test. **p<0.01; ****p<0.0001. Abbreviations are: Monos, monocytes; Macs, macrophages; Alv Macs, alveolar macrophages; Eos, eosinophils.

The online version of this article includes the following figure supplement(s) for figure 9:

**Figure supplement 1.** Inflammatory CC chemokine receptor (iCCR) reporters visualised in resting and inflamed lungs.

**Figure supplement 2.** Reporter co-expression in inflamed lung.

upregulation of CCR1 and CCR5. After 9 days in culture, monocytes were fully differentiated into moDCs and expressed high levels of CCR5. This finding was reinforced by our in vivo studies using adoptive transfer of CCR2+ve monocytes into inflamed air-pouches. We propose that CCR1 and CCR5 are predominantly involved in intra-tissue migration and not recruitment from the circulation.

Under inflammatory conditions, BM and circulating Ly6C$^{hi}$ monocytes still express CCR2. However, the fraction co-expressing CCR1 increases. We reasoned that this may be a consequence of elevated systemic inflammatory cytokines and indeed sustained increase in systemic IFNγ levels recapitulated this phenotype. As the iCCRs, and other chemokine receptors, are transcriptionally regulated in response to a variety of cytokines, this raises the possibility that report of multiple chemokine receptor expression by inflammatory cells in patients with immune and inflammatory disorders is not a reflection of homeostasis but a response to the systemically inflamed environment in these patients. We speculate that this may provide alternative options for leukocyte recruitment from the periphery to inflamed sites and contribute to the failure of specific receptor-targeting therapeutics in inflammatory disease. This hypothesis is supported by our previous findings showing that, in the absence of CCR2, a small proportion of Ly6C$^{hi}$ monocytes can still infiltrate into inflamed tissues (*Dyer et al., 2019*).

After infiltration into the inflamed site, Ly6C$^{hi}$ monocytes still express CCR2 and eosinophils CCR3. Recently infiltrated Ly6C$^{hi}$ monocytes also rapidly increase CCR1 expression, which is maintained after differentiation into F480$^+$ macrophages. This suggests that, in an inflamed context, CCR1 has a pivotal role in directing intra-tissue migration of monocytes and macrophages towards the focus of inflammation and supports the hypothesis that circulating monocytes co-expressing CCR1 and CCR2 might have a recruitment advantage over CCR2-only monocytes. This is further supported by the fact that alveolar macrophages in lung also upregulate CCR1 expression under inflamed conditions, whereas this population does not express any of the iCCRs under resting conditions.

Interestingly, our analyses show that CCR5 is preferentially expressed by macrophages in resting tissues, whereas CCR1 is strongly associated with macrophages recently recruited to inflamed tissues. This could be explained by a hierarchical model of iCCR expression, where BM and circulating Ly6C$^{hi}$ monocytes express CCR2 but, immediately after extravasation, they downregulate CCR2 and upregulate CCR1 as they differentiate into F480$^+$ macrophages. CCR1 would direct macrophages in the early stages of intra-tissue migration. Finally, F480$^+$ macrophages would induce expression of CCR5 and slowly downregulate CCR1 as they fully differentiate into mature macrophages. This model would explain the absence of macrophages expressing exclusively CCR1 in resting tissues, whereas a large proportion of them are CCR1$^+$CCR5$^+$ or CCR5$^+$ only. Alternatively, CCR1 and CCR5 might be expressed in response to different inflammatory stimuli or have different functions in the inflamed tissue. We used carrageenan and LPS to induce inflammation in our study, both signalling through toll-like receptor 4 (*Myers et al., 2019*; *Solov'eva et al., 2013*). This could explain their similar responses, mediated by CCR1$^+$ macrophages. However, CCR5 has been reported to be essential for macrophage recruitment to virus-infected tissues (*Glass et al., 2005*), raising the possibility that alternative inflammatory stimuli could trigger differential responses.

Finally, we failed to detect expression of any of the iCCRs in neutrophils either at rest or during inflammation. We analysed three different founder lines for the iCCR-REP mice and obtained identical results with all of them. This contrasts with previous reports (*Fujimura et al., 2015*; *Gao et al., 1997*; *Gerard et al., 1997*) indicating roles for CCR1 in neutrophil recruitment, but is supported by our findings with iCCR-deficient mice, that show that absence of these chemokine receptors does not lead to deficiencies in neutrophil recruitment to resting or inflamed tissues (*Dyer et al., 2019*). It is possible that differences in inflammatory models used in the various studies, or animal housing arrangements, may have contributed to this apparent disagreement with the literature.

In addition to their use in identifying patterns of iCCR expression in select leukocyte subsets by flow cytometry, these mice can also be used to directly examine iCCR expression in tissue sections, and in living tissues in real time. Importantly, the reporters in these mice are compatible with further use of up to three additional reporters typically associated with the recently developed CODEX technology (*Black et al., 2021*). This will allow investigators to assign phenotypes to iCCR-expressing cells in tissue sections. Clearly the use of four fluorescence channels, to detect the reporters, limits the

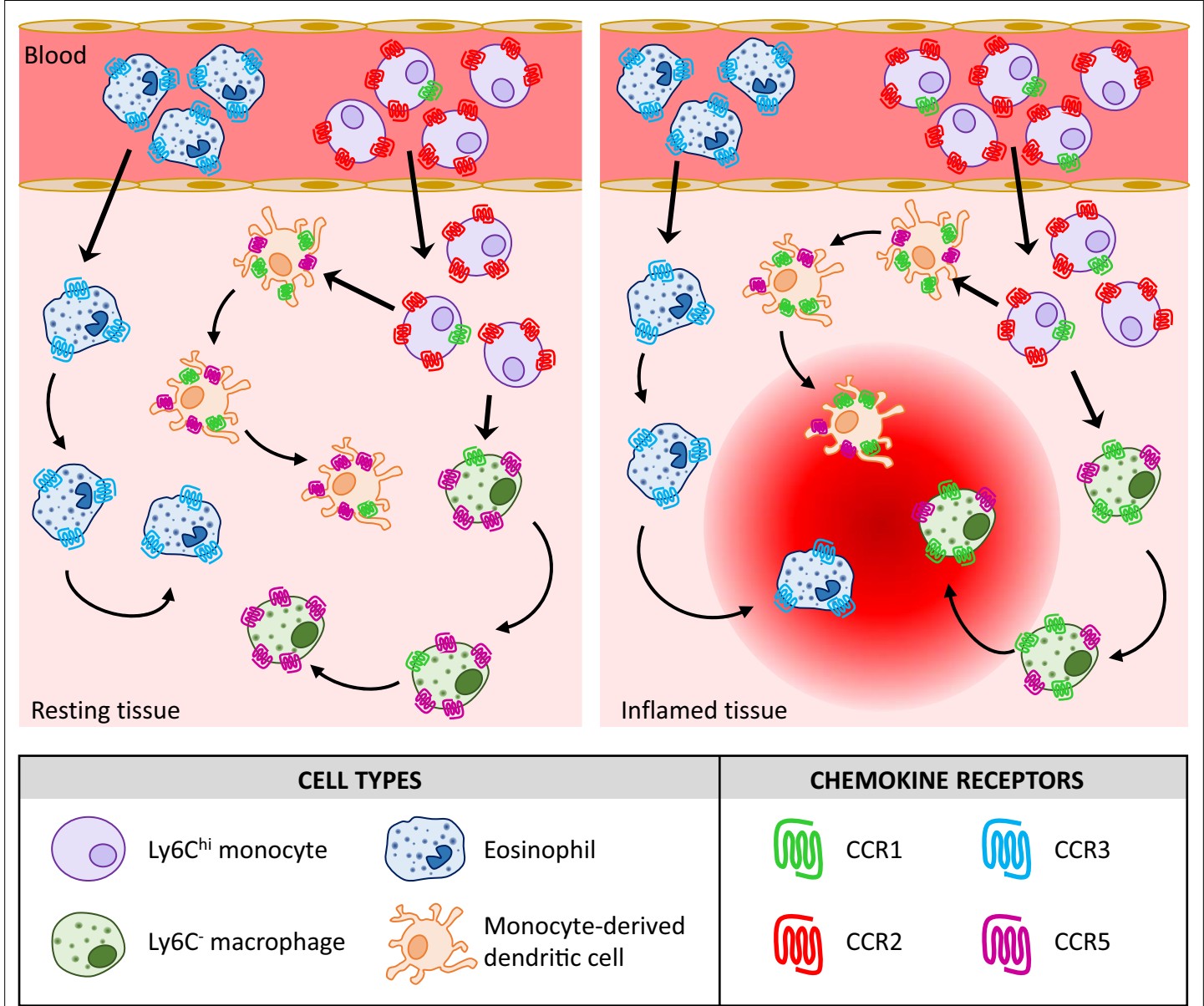

**Figure 10.** Proposed model for the contribution of the inflammatory CC chemokine receptors (iCCRs) to leukocyte migration.

ability to precisely define cell phenotypes in in vivo real-time imaging but this can be, to some extent, alleviated by adoptive transfer of isolated reporter-expressing leukocyte subpopulations.

In summary, therefore, our analyses indicate that the previous belief that individual inflammatory leukocytes express multiple iCCRs (*Haringman et al., 2006*; *Tacke et al., 2007*; *Weber et al., 2000*) is probably wrong and that iCCR expression is more specific (summarised in *Figure 10*) than previously believed. For example, it was previously reported that monocytes simultaneously express CCR1, CCR2, and CCR5 but our data indicate that, at least for classical monocytes, they are almost exclusively singly positive for CCR2 amongst the iCCRs. Overall our data suggest non-redundant roles for the iCCRs in leukocyte cell trafficking at rest and in acute inflammation. We propose that the iCCR-REP mice represent a transformational technical advance permitting in-depth analysis of receptor expression in a range of resting and pathological conditions that will shed important light on chemokine receptor biology and potentially inform future rational drug design.

## Materials and methods

### Mouse generation and maintenance

The iCCR-REP mice were generated in collaboration with Taconic Biosciences. First, the inflammatory *Ccr* genes were targeted in a BAC using counterselection recombineering as previously described (*Wang et al., 2014*) in order to replace the coding sequence of each one of the inflammatory *Ccr* genes with the sequence coding for a different fluorescent reporter. Then, the iCCR-REP cluster was incorporated into the mouse genome using pro-nuclear injection, thus generating transgenic iCCR-REP mice. The presence of the iCCR-REP cluster in these animals was confirmed by PCR using primers specific for the iCCR reporters (*Table 1A*). Quantification of the number of copies of the iCCR-REP cluster inserted into the mouse genome was done by QPCR with the primers listed in *Table 1B*, using the TBP gene as a reference, because it sits outside of the iCCR cluster and remained unaltered in the process. Finally, the iCCR-REP cluster was localised to chromosome 16:82389380– 82392016 using TLA as described elsewhere (*de Vree et al., 2014*). iCCR-def mice were generated in-house (*Dyer et al., 2019*) and CCR5-def mice were a kind gift from Dr Takanori Kitamura, University of Edinburgh. All animal strains were generated and maintained on a C57BL/6 background and were bred in a specific pathogen-free environment in the animal facility of the Beatson Institute for Cancer Research (Glasgow). Routine genotyping of all animals was done by PCR of ear samples (Transnetyx). All experiments were done on animals 10–12 weeks of age, using congenic WT animals derived from heterozygous crosses as controls. All experiments were carried out under the auspices of a UK Home Office Project License and were approved by the local Ethical Review Committee.

### Resting tissue isolation

Resting mice were sacrificed and tissues extracted for flow cytometry or RNA analysis.

Blood was extracted from the vena cava and mixed with 100 µL of 0.5 M EDTA (Thermo Fisher Scientific) prior to red blood cell lysis using an ACK lysis solution (Thermo Fisher Scientific) as per manufacturer's instructions. Leukocyte content was then used for further analysis. Immediately after blood isolation, mice were perfused using 20 mL of PBS (Thermo Fisher Scientific) containing 2 mM EDTA (Thermo Fisher Scientific).

BM was then extracted from the femur and tibia of the mice, red blood cells lysed as described above, and leukocyte content used in downstream analyses.

Spleen, lungs, and kidneys were isolated from perfused animals and incubated overnight (O/N) in RNAlater stabilisation solution (Thermo Fisher Scientific) for RNA analysis. Alternatively, these tissues were processed for antibody staining and flow cytometry analysis. Dissected spleens were filtered through 70 µm nylon mesh membranes, washed with PBS, and red blood cells lysed as described above, prior to antibody staining. Lungs were cut into small pieces and digested in 5 mL of RPMI containing DNase I (100 µg/mL, Roche), Dispase II (800 µg /mL, Roche), and Collagenase P (200 µg/ mL, Roche) for 90 min at 37°C. Kidneys were cut into small pieces and digested in 4 mL of PBS containing calcium and magnesium, HEPES (20 mM, VWR), collagenase I (1.8 mg/mL, Sigma-Aldrich), collagenase XI (156 µg/mL, Sigma-Aldrich), and Hyaluronidase (158 µg/mL, Sigma-Aldrich) for 20 min at 37°C. After digestion, enzymes were deactivated using 20 µL of foetal bovine serum (FBS) and lung

**Table 1.** Primers used in the study.

A. Detection of the iCCR reporters

| | |
|---|---|
| iRFP682 QPCR1 | GTCACCCCAGACCTCAATCC |
| iRFP682 QPCR2 | AACGATCAATCCCCACAGTC |
| mRuby2 RT F | TGGGAAAGAGTTACGAGATACGA |
| mRuby2 RT R | AACGAGACAGCCATCCTCAA |
| Clover QPCR1 | AACGGCATCAAGGCTAACTTC |
| Clover QPCR2 | GGGTGTTCTGCTGGTAGTGG |
| mTagBFP2 QPCR1 | ACCGTGGACAACCATCACTT |
| mTagBFP2 QPCR2 | CCTCGACCACCTTGATTCTC |

B. Quantification of iCCR-REP cluster insertion

| | |
|---|---|
| CCR1prom QPCR1 | TCAACTCAACTCCATCCAACC |
| CCR1prom QPCR2 | CTGTCTTTCCTCTCTGCTCCA |
| BAC-CPN1 Stan1 | CAGCTAGCCCCCAGGTGACA |
| BAC-CPN1 Stan2 | AGTCTTTCTTTCCTGCGTTGTATG |
| BAC-CPN1 QPCR1 | GATAAAGGGAAGCAGACACCAG |
| BAC-CPN1 QPCR2 | CAGCAGGGAGGAAAGAAGAGT |
| BAC-CPN2 Stan1 | AGTCTTTCTTTCCTGCGTTGTATG |
| BAC-CPN2 Stan2 | AAAACCAGACAGGATAGATAACTG |
| BAC-CPN2 QPCR1 | AGGGGTGGAAGCCTATCTCTAC |
| BAC-CPN2 QPCR2 | TGGCAGCATTTACAGGGTCT |
| BAC-CPN3 Stan1 | GGATGGGAGGGAATTTGGAGAAGA |
| BAC-CPN3 Stan2 | GCTTTGTGAAGGCCGAGGTCTAA |
| BAC-CPN3 QPCR1 | CCCCATCCATAACACAAACC |
| BAC-CPN3 QPCR2 | CAAAATGAGCACCTCCCTTC |
| CCR2exon Stan1 | AGGGAGAGCAGAAGGCTAA |
| CCR2exon Stan2 | CCCAGGAAGAGGTTGAGAGA |
| CCR2exon QPCR1 | TGTGGGACAGAGGAAGTGG |
| CCR2exon QPCR2 | GGAGGCAGAAAATAGCAGCA |
| CCR5exon Stan1 | ACCCATTGAGGAAACAGCAA |
| CCR5exon Stan2 | CTTCTGAGGGGCACAACAAC |
| CCR5exon QPCR1 | TTTGTTCCTGCCTTCAGACC |
| CCR5exon QPCR2 | TTGGTGCTCTTTCCTCATCTC |
| TBP Stan1 | GAGTTGCTTGCTCTGTGCTG |
| TBP Stan2 | ATACTGGGAAGGCGGAATGT |
| TBP QPCR1 | TGCTGTTGGTGATTGTTGGT |
| TBP QPCR2 | AACTGGCTTGTGTGGGAAAG |

and kidney cell suspensions were filtered through 70 µm nylon mesh membranes, washed with PBS, and stained for flow cytometry analysis.

## Air-pouch model

The air-pouch model of inflammation was used as previously described (*Dyer et al., 2019*). In brief, 3 mL of sterile air were injected under the dorsal skin on three occasions over a period of 6 days to induce the formation of a subcutaneous air cavity. One mL of sterile carrageenan (1% (w/v) in PBS, Sigma-Aldrich) was inoculated into the cavity to induce inflammation 24 hr after the last air injection. Forty-eight hours later, mice were culled, and blood and BM samples were collected as detailed above. The cavity was flushed with 3 mL of PBS containing 2 mM EDTA and 2% (v/v) FBS and the lavage fluid was collected, washed with PBS, and stained for flow cytometry analysis. The membrane surrounding the air-pouch was then isolated and digested in 1 mL of Hanks buffered saline solution (Thermo Fisher Scientific) containing 0.44 Wünsch units of Liberase (Roche) for 1 hr at 37°C and 1000 rpm shaking. After digestion, Liberase was neutralised using 20 µL of FBS and cell suspensions were filtered through 70 µm nylon mesh membranes, washed with PBS, and stained for flow cytometry analysis.

For the transfer of REP monocytes into the inflamed air-pouch, $8.5 \times 10^5$ mRuby2/CCR2$^+$ monocytes isolated from resting BM were injected into the air-pouch of WT animals. Mice were culled 72 hr later and air-pouch tissues were collected for leukocyte analysis as described above.

## LPS model of lung inflammation

For the induction of lung inflammation using *Escherichia coli* LPS, mice were anaesthetised using inhaled isoflurane (4% (v/v) isoflurane and 2 L $O_2$/min) and 30 µL of LPS (250 µg/mL, Sigma-Aldrich) or vehicle PBS were administered intranasally. Forty-eight hours later, mice were culled, perfused, and lungs were isolated for flow cytometry analysis as detailed above.

## Implantation of IFNγ-loaded subcutaneous osmotic pumps

Mice were anaesthetised using inhaled isoflurane (2% (v/v) isoflurane and 2 L $O_2$/min) and maintained under these conditions during the surgical procedure. A small pocket was generated under the dorsal skin, where IFNγ- (100 ng/µL) or vehicle PBS-loaded osmotic pumps (Alzet osmotic pumps, model 2001; Charles River) were implanted. Infusion of IFNγ (100 ng/hr) or PBS was maintained for 7 days. After this time, animals were sacrificed and BM and blood were extracted for flow cytometry analysis as described above.

**Table 2.** Antibodies used in the study.

| Antibody | Clone | Source | Working dilution |
| --- | --- | --- | --- |
| Anti-mouse CD45 | 30-F11 | eBioscience | 1/100 |
| Anti-mouse CD11b | M1/70 | eBioscience | 1/100 |
| Anti-mouse SiglecF | E50-2440 | BD Bioscience | 1/100 |
| Anti-mouse F4/80 | BM8 | eBioscience | 1/100 |
| Anti-mouse CD64 | X54-5/7.1 | BD Bioscience | 1/100 |
| Anti-mouse Ly6C | HK1.4 | BioLegend | 1/100 |
| Anti-mouse CD11c | HL3 | BD Bioscience | 1/100 |
| Anti-mouse MHCII | M5/114.15.2 | BioLegend | 1/100 |
| Anti-mouse Ly6G | 1A8 | BD Bioscience | 1/100 |
| Anti-mouse CD19 | eBio1D3 (1D3) | eBioscience | 1/100 |
| Anti-mouse CCR2 | SA203G11 | BioLegend | 1/50 |
| Anti-mouse CCR3 | J073E5 | BioLegend | 1/50 |
| Anti-mouse CCR5 | HM-CCR5 (7A4) | eBioscience | 1/50 |

## Flow cytometry

Tissue lysates were prepared as described above and stained for 20 min at 4°C with 100 μL of fixable viability stain (eBioscience). Cells were then washed in FACS buffer (PBS containing 2 mM EDTA and 2% (v/v) FBS) and stained for 20 min at 4°C with 50 μL of antibody cocktail containing subset-specific antibodies (*Table 2*) diluted in Brilliant Stain Buffer (BD Biosciences). Cells were washed in FACS buffer and fixed for 20 min at 4°C in 100 μL of Fixation Buffer (BioLegend).

Stained samples were analysed on a BD LSRFortessa flow cytometer (BD Biosciences) and data analysis was performed using FlowJo software (FlowJo).

For the isolation of REP monocytes from BM for transfer experiments, cells were isolated and stained as described above. After staining, cells were analysed on a FACS Aria sorter (BD Biosciences) and resuspended in GM-CSF containing media ready for injection into the inflamed air-pouch.

## Generation of BM-derived moDCs

BM cells were isolated as described above and $10^7$ cells were resuspended in 10 mL of RPMI-1640 (Sigma) supplemented with FBS (10% v/v), L-glutamine (1% v/v), 50 μM β-mercaptoethanol, primocin, and 20 ng/mL of murine recombinant GM-CSF (Peprotech). Cells were transferred to tissue culture-treated Petri dishes. At days 2 and 9, the medium was collected, centrifuged at 300 $g$ for 5 min, and pelleted cells were analysed for the expression of iCCR reporters by flow cytometry. Monocytes and moDCs were identified on the basis of Ly6C, F480, CD11c, and MHCII expression (monocytes were CD11c$^-$ F480$^+$ Ly6C$^{hi}$; moDC precursors were F480$^+$ CD11c$^+$ Ly6C$^-$; mature moDCs were CD11c$^+$ MHCII$^{hi}$).

## RNAscope for the detection of CCR1 mRNA

Kidneys were isolated from resting mice after PBS perfusion and fixed O/N in 20 mL of 10% Neutral Buffered Formalin (Leica). Fixed kidneys were paraffin-embedded using a Shandon Citadel 1000 tissue processor (Thermo Fisher Scientific) and stored at room temperature (RT) until used. The day prior to CCR1 mRNA analysis, kidneys were wax-embedded and sliced into 6 μm sections onto SuperFrost Plus adhesion slides (Thermo Fisher Scientific). Slides were air-dried O/N at RT and, the following day, CCR1 mRNA was detected using an RNAscope target probe specific for the gene with the RNAscope 2.5 HD Reagent Kit-RED (Advanced Cell Diagnostics). Manufacturer's instructions were followed, with minor modifications. Specifically, kidney sections were incubated in the RNAscope Target Retrieval Reagent for 18 min and were treated with the RNAscope Protease Plus Reagent for 35 min. Images were acquired on an Evos FL Auto 2 microscope (Thermo Fisher Scientific).

## Imaging

Lungs, spleens, and kidneys were isolated from resting mice after PBS perfusion. They were immersed in 4 mL of 4% paraformaldehyde (VWR), incubated at RT for 1 hr, and finally incubated O/N at 4°C to achieve full fixation. Tissues were then immersed in increasing concentrations of sucrose (10%, 20%, and 30% (w/v) in PBS; Fisher Chemicals) for 18–24 hr in each solution. Finally, tissues were frozen in O.C.T. embedding media (Tissue-Tek) and stored at –80°C; 48–24 hr prior to sectioning, tissues were transferred to –20°C. After –20°C incubation, tissues were sliced into 8 μm sections onto SuperFrost Plus adhesion slides and transferred to –20°C again until processed for imaging.

The day of imaging, tissue sections were incubated at 60°C for 10 min and washed twice in PBS for 5 min. Slides were then incubated for 20 min in 0.1 M glycine (VWR) with mild shaking to reduce background autofluorescence. After glycine incubation, sections were washed five times with PBS for 5 min and then immersed in cold water until mounted using 10 μL of Mowiol mounting medium. Tissues were imaged using an Axio Imager M2 microscope (Zeiss) or a spinning disk Axio Observer Z1 confocal microscope (Zeiss).

High background autofluorescence in *Figure 5Bvii* was removed using proprietary image clearing software. Sparca (https://www.sparca.com), headquartered in London, is a bio-inspired mixed signal processing company working in the medical, scientific, and electronic consumable space. For this project, Sparca applied a number of proprietary image signal processing algorithms which allowed the identification of the main particles of interest and the minimisation of the background. Furthermore, another set of algorithms was applied to bring to light some of the cells which were not fully visible to the naked eye in the original image.

Live cell imaging was carried out using a spinning disc confocal microscope (Zeiss) with a heated stage, at 37°C, 5% $CO_2$. Live mammary glands were visualised in glass bottom dishes (Nunc, Thermo), in PBS containing 5% FBS (Sigma).

## Quantification and statistical analysis

All analyses were performed using the Prism software package (GraphPad). Normally distributed data were analysed using unpaired t-test with or without Welch's correction, according to their standard deviations. Not normally distributed data were analysed using Mann-Whitney or Kolmogorov-Smirnov, according to their standard deviations. In all analysis, $p=0.05$ was considered the limit for statistical significance.

## Acknowledgements

We acknowledge the assistance of the Institute of Infection, Immunity and Inflammation Flow Core Facility at the University of Glasgow. We also thank Dr Michael Z Lin and Prof. Vladislav V Verkhusha for their advice on the selection of the fluorescent reporters.

## Additional information

### Funding

| Funder | Grant reference number | Author |
|---|---|---|
| Wellcome Trust | 217093/Z/19/Z | Laura Medina-Ruiz<br>Robin Bartolini<br>Douglas P Dyer<br>Francesca Vidler<br>Catherine E Hughes<br>Fabian Schuette<br>Marieke Pingen<br>Gerard J Graham |
| Medical Research Council | MRV0109721 | Samantha Love<br>Marieke Pingen<br>Alan James Hayes<br>Gerard J Graham |
| Max-Planck-Institute for Cell Biology and Genetics | open access funding | Adrian Francis Stewart |
| University of Glasgow | | Gerard J Graham |

The funders had no role in study design, data collection and interpretation, or the decision to submit the work for publication. For the purpose of Open Access, the authors have applied a CC BY public copyright license to any Author Accepted Manuscript version arising from this submission.

### Author contributions

Laura Medina-Ruiz, Conceptualization, Data curation, Formal analysis, Investigation, Methodology, Project administration, Supervision, Writing – original draft, Writing - review and editing; Robin Bartolini, Douglas P Dyer, Conceptualization, Data curation, Formal analysis, Investigation, Supervision, Writing – original draft; Gillian J Wilson, Formal analysis, Investigation, Methodology; Francesca Vidler, Samantha Love, Data curation, Formal analysis, Investigation, Writing – original draft; Catherine E Hughes, Data curation, Formal analysis, Project administration, Writing – original draft; Fabian Schuette, Conceptualization, Formal analysis, Investigation, Writing – original draft; Marieke Pingen, Alan James Hayes, Data curation, Investigation, Methodology; Jun Fu, Conceptualization, Investigation, Methodology, Writing – original draft; Adrian Francis Stewart, Conceptualization, Formal analysis, Methodology, Supervision, Writing – original draft; Gerard J Graham, Conceptualization, Data curation, Formal analysis, Funding acquisition, Investigation, Project administration, Supervision, Writing – original draft, Writing - review and editing

Author ORCIDs
Laura Medina-Ruiz http://orcid.org/0000-0002-2934-534X
Alan James Hayes http://orcid.org/0000-0003-2708-6230
Adrian Francis Stewart http://orcid.org/0000-0002-4754-1707
Gerard J Graham http://orcid.org/0000-0002-7801-204X

Ethics
All experiments were carried out under the auspices of a UK Home Office Project License and were approved by the local University of Glasgow Ethical Review Committee.

Decision letter and Author response
Decision letter https://doi.org/10.7554/eLife.72418.sa1
Author response https://doi.org/10.7554/eLife.72418.sa2

## Additional files

### Supplementary files
• Transparent reporting form

### Data availability
Data relating to this study are available on Dryad (https://doi.org/10.5061/dryad.3r2280gjs). Mouse lines generated in this study will be available, on request, from the corresponding author.

The following dataset was generated:

| Author(s) | Year | Dataset title | Dataset URL | Database and Identifier |
|---|---|---|---|---|
| Graham GJ, Medina-Ruiz L, Bartolini R, Wilson G, Dyer D, Vidler F, Hughes C, Schuette F, Love S, Pingen M, Hayes A, Fu J, Stewart F | 2022 | Data for ELIfe paper entitled 'Analysis of combinatorial chemokine receptor expression dynamics using multi-receptor reporter mice | https://doi.org/10.5061/dryad.3r2280gjs | Dryad Digital Repository, 10.5061/dryad.3r2280gjs |

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
