## [Editor Report]

The manuscript describes the potential of unique multi-chemokine receptor reporter mice to track CCR expression dynamics in myeloid cells in steady state and inflammatory conditions which may overcome some existing limitations to finding accurate antibodies for these receptors. This paper will be of broad interest to investigators studying the role of chemokines and their receptors in animal models of human disease. The conclusions of the authors are generally well-supported by the data and provide novel and important insights into the hierarchy of chemokine receptor expression during the recruitment of myeloid cells to sites of inflammation.

---

## [Decision Letter]

**Decision letter after peer review:**

Thank you for sending your article entitled "Analysis of combinatorial chemokine receptor expression dynamics using multi-receptor reporter mice" for peer review at *eLife*. Your article is being evaluated by 2 peer reviewers, and the evaluation is being overseen by a Reviewing Editor and Satyajit Rath as the Senior Editor.

We all recognized that the mouse model is very interesting but its unique advantages have not been adequately demonstrated and not leveraged for unique insights. Thus, we would like to evaluate a plan of action to demonstrate its unique advantages, with rigorous and detailed data, before a final decision can be made.

It is important for the authors to demonstrate that the novel reporter mice allow studies of chemokine receptor expression during cell recruitment that could not reasonably be done without their novel strain. This would require properly controlled studies (including imaging studies), presentation of MFI and/or cell numbers (not just percentages of positive cells). Indeed, imaging approach is the true interest of the mouse and this was not harnessed. No quantification was provided, and the images are not convincing, lot of autofluorescence in the kidney for instance.

It would also be good to examine cell types besides myeloid cells. For each of their models, the authors should indicate which cell types are actually recruited to the sites of inflammation (ie no significant changes from steady state), and if MFI of receptor expression correlates with the recruitment. If important cell types are not recruited, the authors should at least acknowledge the limitations of the models they have chosen.

All chemokine receptors except CCR1 can be monitored using antibodies by flow, so relying on fluorescent reporter is likely less informative than the actual protein expression. We share this problem that the authors claim they are tracking CKR expression while in fact they follow reporters. It is very different regarding the conclusions made in the manuscript.

Provide absolute number quantification vs % to support conclusions made.

Variation in MFI can maybe be added when discussing about the level of CKR expression.

*Reviewer #1 (Recommendations for the authors):*

The manuscript introduces a new transgenic mouse strain with a combination of four CC-chemokine receptor fluorescent reporters. This tool represents a very nice approach to study the complexity of inflammatory chemokine receptor network in the orchestration of leukocyte mobilization and interaction. Monitoring CCR expression is challenging due to important non-specific staining, cross reactivity or even absence of functional antibody. Fluorescent reporter mice are indeed very fruitful tools to track cells by imaging or flow cytometry albeit one has to keep in mind that they reflect mostly mRNA expression which physiological expression is often impacted by genomic modifications, hence the true relationship between fluorescent reporter expression and the actual protein needs accurate attention to draw functional conclusions. The present manuscript does not evaluate sufficiently these limitations and tend to overinterpretation.

The main innovation of the strain relies on its potential for imaging studies. It is the first opportunity to map iCCR+ cell distribution in order to provide insights in the organization of the iCCR network but this aspect is unfortunately barely exploited and the few provided images do not support the model. The image in the spleen is very nice but what is the information generated from it? As the authors support the originality of their transgene to study the dynamic of iCCR expression, one could have expected to study the pattern of iCCR+ cell distribution in steady state vs inflammatory condition. Flow analysis could have been performed using specific antibody (controlled with the iCCR-def mouse) with indeed the exception of CCR1. If it is important to generate a reporter for Ccr1, the paper does not harness the added-value of the other reporters.

More specific points

The authors focus their analysis on myeloid cells exclusively and address the relative expression of the fluorescent reporters on the gated subsets.

As the mouse lack of previous validation, it is important to better characterize the system. The authors should deeply analyse all the leukocyte expressing the different fluo. For instance, in the Ccr2-RFP model, NK, T cell and Treg do express the RFP. One could expect CCR1 and CCR5 reporters to be expressed as well in lymphocytes, and what would be the relative proportion of fluorescent monocyte and macrophage among all. This is particularly important to apprehend the imaging part.

The provided images lack of quantification and control for specificity. Considering that other cells than monocyte and macrophages might express the fluorescent reporters, it is unclear what the image really show.

Renal tubules are very auto fluorescent and seems to be the main source of signal in Figure 5Bvii.

The authors correlate fluorescent reporter to the CK receptor expression using specific anti-CCR2, anti-CCR3 and anti-CCR5 on splenic cells to justify their conclusion of Figure 2. This claim needs reconsideration as the authors show only in the spleen, only on selected populations (Mo for CCR2, eosinophils for CCR3 and kidney Mac for CCR5) and only at steady state. Moreover, this claim is extended to CCR1 reporter for which protein expression cannot be validated.

The receptor /reporter correlation analysis already show that the reporter MFI does not strictly correlate with the antibody labeling MFI. However, the authors make assumptions all along the manuscript that they can track the dynamic of CKR expression and draw mostly their conclusion based on the percentage of iCCR+ cells. The graphical abstract suggests that the CKR expressions per cell are modified in inflammatory conditions but does not reflect the change in cell frequency as supported by the data.

Overall more extensive characterization of this correlation in the different cell subsets for each receptor, tissues and experimental conditions need to be provided and the conclusions need to be redrawn accurately all across the manuscript.

Beyond the points raised in the public review that need to be addressed I have a set of specific recommendations below:

In the lung the gating strategy does not allow to capture non-classical monocytes which are more abundant than Ly6Chigh Mo and interstitial macrophages at steady state. Adding a CD43 staining would address that. This is a limit for the LPS model.

Non-classical Mo should be expected to express CCR5 (from human studies at least). Is it confirmed by the multiple reporter mouse?

All across the manuscript the authors need to refer to the reporter expression and not the CCR protein whenever they use the fluorescence.

Changes in % of iCCR+ cells do not reflect any down or upregulation of the receptor. For instance, line 336, the authors evaluate a 6-fold increase in CCR1 expression after IFNγ treatment. It seems that the frequency of Clover+ cells is increased which does not necessarily reflect increased CCR1 expression but accumulation of Clover+ cells or reduction of Clover- cells. Comparative MFI on iCCR+ subset in the different conditions along with the % of iCCR+ cells might reflect at least, change in the activation state of the cell. All statements regarding CCR expression should be carefully considered in the manuscript.

Naming F4/80+ macrophages in the lung is unclear as AM are excluded from the analysis. The authors could consider using F4/80+ interstitial macrophage for clarification.

It appears that bimodal level of fluorescent reporter expression can be detected sometimes (CCR5, CCR1 in lung mac, CCR2 in Mo from air pouch membrane, CCR3 in air pouch eosinophils). It suggests that the authors deal with distinct subsets of cells which are considered as one. This need clarification

CCR2 MFI in kidney and lung Mo is bimodal (Figure 4-5Bvi). can the author explain why?

Dot plots showing co-expression of the reporters (as shown only FigS4Ci, ii) along with the pie charts would be more convincing and visual for the reader rather than the individual expression in all figures. Moreover, it will more clearly show potential spectral overlap between the different reporters.

The assumption that Monocytes downregulate Ccr2 and upregulate Ccr5 and Ccr1 upon mac differentiation in vitro is fair. But this conclusion cannot be extended to the in vivo data as proposed. Adoptive transfer experiment of purified Mo should be performed to confirm this claim.

*Reviewer #2 (Recommendations for the authors):*

Medina-Ruiz and colleagues have generated a transgenic mouse line that should prove helpful in unraveling the complex biology of chemokines and their receptors. Previous studies in this area have been hampered by the high background seen with antibodies against chemokine receptors, particularly CCR1. The mouse line generated by the authors contains a bacterial artificial chromosome (BAC) in which the coding regions of four inflammatory chemokine receptors were replaced by coding regions of different fluorescent reporters. Importantly, the authors show that the reporters for the various chemokine receptors accurately mimic results generated using antibodies against those same receptors. The genetic approach is elegant, and the BAC transgenic mouse line will be very useful to investigators seeking to understand chemokine biology in different settings. In the current manuscript, the authors characterize their novel mouse strain in two different models of inflammation: the air pouch model, and an inhaled LPS-induced model of pulmonary inflammation. Analysis of the transgenic mouse strain in these two models has revealed novel information regarding the spatio-temporal expression of these receptors during inflammation. For example, the authors show that monocytes express CCR2, but as these cells differentiate into macrophages, they tend to lose CCR2 expression and instead begin to express CCR1 and/or CCR5. The latter receptor appears to be dominant in resting macrophages at steady state. The study therefore provides critical insight into the hierarchical relationship of chemokine receptor expression during the initiation and resolution of inflammation.

The paper is very well written, the experiments appear to have been performed well, and the authors' claims and conclusions regarding chemokine receptor display are largely justified by their data. However, some caution is warranted regarding conclusions about recruitment of certain cell types. For example, although the authors show that CCR3 is the only receptor expressed on circulating eosinophils, these cells are not recruited in significant numbers to the air pouch or lungs of LPS-treated mice. Also, because numbers of these cells at steady state are presented in a different figure, it is difficult to assess 'recruitment' of eosinophils in these models. Other chemokine receptors, including CCR5, have been reported to be expressed on eosinophils and direct chemotaxis of these cells. Thus, it will be helpful to know if other receptors besides CCR3 are expressed on eosinophils in animal models of asthma or atopic dermatitis, where these cells are a major component of the inflammatory response.

The authors do a good job of evaluating chemokine reporter expression in lung interstitial macrophages, which express the markers CD11b and F4/80. However, their analysis of conventional dendritic cells (cDC) is limited. cDCs are critically important cells for generating adaptive immune responses, and in the lung comprise two major populations, DC1 and DC2. Going forward, it will be interesting to know which chemokine receptors are expressed in these cell types, and when. Nonetheless, establishment of this important mouse cell line is an important step towards an improved understanding of how hierarchical expression of different inflammatory chemokine receptors function to recruit DCs (and their precursors) in models of human disease.

Macrophages and DCs in the lung share many surface molecules but they can be distinguished by display of molecules such as F4/80 and CD88. On lines 264-265, the authors state that, "co-expression of CCR1 or CCR5 with CCR2 was more apparent in F480LoMHCIIHi than in F480MHCIIHi + IMs (Figure S4Ci-iv). I have two questions. First, in the latter cells, was the "+" supposed to refer to F4/80 levels? The authors seem to have accidentally left out the superscripted indicator of expression for that marker. Second, why do the authors call the F480loMHChi cells 'interstitial macrophages', rather than conventional (c)DCs, specifically DC2? The data could be interpreted to mean that cDC2s maintain CCR2 longer than do macrophages, which would be an interesting observation.

There is no mention of lung CD103+ DCs (DC1), which can efficiently cross-present antigens to naïve T cells and have an important role is activation of CD8^+^ T cells. In addition to their display of CD103, DC1 can also be distinguished from DC2 by relatively low levels of CD11b. Thus, they can be identified as CD11chiMHCII hiCD103+CD11blow cells. However, I understand if the authors feel that this more detailed study is beyond the scope of the current paper.

---

## [Author Response]

Reviewer #1 (Recommendations for the authors):The manuscript introduces a new transgenic mouse strain with a combination of four CC-chemokine receptor fluorescent reporters. This tool represents a very nice approach to study the complexity of inflammatory chemokine receptor network in the orchestration of leukocyte mobilization and interaction. Monitoring CCR expression is challenging due to important non-specific staining, cross reactivity or even absence of functional antibody. Fluorescent reporter mice are indeed very fruitful tools to track cells by imaging or flow cytometry albeit one has to keep in mind that they reflect mostly mRNA expression which physiological expression is often impacted by genomic modifications, hence the true relationship between fluorescent reporter expression and the actual protein needs accurate attention to draw functional conclusions. The present manuscript does not evaluate sufficiently these limitations and tend to overinterpretation.

The reviewer is of course right that reporters generally reflect transcription and that their ability to properly report protein expression is affected by numerous factors including reporter half-life and potential post-transcriptional modification events. We are indeed aware of the limitations of the use of reporter strains. However, we have gone to some lengths to show that, in the current mouse model, reporter expression faithfully recapitulates iCCR surface presentation. We demonstrate this in vivo (Figure 2 of our manuscript) and further show that reporter expression is dynamic, changing, for example, as monocytes differentiate (Figures 3-5 of the manuscript and new Figure 4—figure supplements 2 and 3) or under different inflammatory conditions (Figures 6-9 of the manuscript). Some of these issues are dealt with in more detail below. In addition, we have now included a further, more critical, discussion of the limitations of our model (pages 15-16 of the revised manuscript) and hope that this is an appropriate way to address the reviewer’s concern.

The main innovation of the strain relies on its potential for imaging studies. It is the first opportunity to map iCCR+ cell distribution in order to provide insights in the organization of the iCCR network but this aspect is unfortunately barely exploited and the few provided images do not support the model. The image in the spleen is very nice but what is the information generated from it?

The reviewer is correct that the spleen image on its own, whilst useful, provides limited information on what specific cell types express the receptors. However, by using the recently described CODEX technology it will be relatively simple to incorporate additional phenotypic markers to identify individual cell types expressing the receptors. We have now included a discussion of this point on page 18 of the revised manuscript and hope that this is sufficient. If, however, the reviewer feels that some CODEX-based co-staining for cell phenotype is essential to prove the utility of the mice for fluorescent imaging, we will certainly try to do this although lab and animal access is still severely limited and this may therefore take some time.

As the authors support the originality of their transgene to study the dynamic of iCCR expression, one could have expected to study the pattern of iCCR+ cell distribution in steady state vs inflammatory condition.

We thank reviewer for this comment and whilst we have presented substantial data on the use of flow cytometry in this regard, we have now included fluorescent images of resting and inflamed lung tissues which we hope will address this concern (Figure 9—figure supplement 1 and pages 13-14 of the revised manuscript).

Flow analysis could have been performed using specific antibody (controlled with the iCCR-def mouse) with indeed the exception of CCR1. If it is important to generate a reporter for Ccr1, the paper does not harness the added-value of the other reporters.

We apologise if we have misinterpreted the reviewer’s comments. However, we do indeed provide such information in Figure 2 of the revised manuscript (see below for a further discussion of this point). We presume that here the reviewer is referring to the comparison of the reporters and the antibodies in detecting receptor expression in models of inflammation and in a variety of resting tissues. There are a number of reasons why our reporter mice represent added value compared to the use of a CCR1 reporter alongside antibodies:

i) Many of the protocols used for tissue separation and cell isolation rely on enzymatic or mechanical disaggregation of tissues, which ultimately damages the structure of the iCCRs on the cell surface, destroying the epitopes detected by the antibodies. In this way, antibodies that detect, for example, CCR3 very efficiently on tissues like blood or air pouch fluid (which do not require enzymatic treatments prior to antibody staining), fail to detect it on tissues which require enzymatic digestion to release leukocytes. This issue is discussed on pages 15-16 of the revised manuscript.

ii) Another general problem with anti-CCR antibodies is lack of specificity. The CCRs (and in particular CCRs 1, 2, 3 and 5) have a high degree of homology, leading to non-specific binding of the antibodies, as we show in Figure 2 (particularly, Figure 2C). Such non-specific binding, obvious when antibody binding is assessed using iCCR KO mice, is frequently not properly controlled for and leads to false positive detection when the antibody is tested using only isotype or FMO (fluorescence minus one) controls. We have now included a short addition to the Discussion section (page 15 of the revised manuscript) of the manuscript addressing this issue. Importantly, we have done extensive analysis of these iCCR detection antibodies which is probably outside the scope of the current manuscript, but we would be very happy to share this information with the reviewer if that would be useful.

iii) Finally in terms of added value, and in contrast to the use of antibodies, these reporter mice can be used to study the real-time in vivo dynamics of receptor expression on either resident, or adoptively transferred, cells. We have now included a video from real-time imaging of receptor expression on leukocytes in the living mammary gland (Video 1 and discussed on page 11 of the revised manuscript). We hope that this further illustrates the added experimental value of the multi-receptor reporter mice.

More specific pointsThe authors focus their analysis on myeloid cells exclusively and address the relative expression of the fluorescent reporters on the gated subsets.As the mouse lack of previous validation, it is important to better characterize the system. The authors should deeply analyse all the leukocyte expressing the different fluo.

As the reviewer points out, it was our express intention to demonstrate the utility of this reporter mouse strain by focusing on myeloid cells and we hope that the data presented adequately achieve this aim. However, we have analysed expression on other cell types including lymphoid cells and now present data relating to reporter expression on subsets of lymphoid cells in new Figure 8—figure supplement 2 and these data are described on page 13 of the revised manuscript.

For instance, in the Ccr2-RFP model, NK, T cell and Treg do express the RFP. One could expect CCR1 and CCR5 reporters to be expressed as well in lymphocytes, and what would be the relative proportion of fluorescent monocyte and macrophage among all. This is particularly important to apprehend the imaging part.

As shown in new Figure 8—figure supplement 2 (and discussed on page 13 of the revised manuscript), we do indeed see strong CCR2 and CCR5 expression on NK cells and CCR2 expression on T cells in the air pouch model. We also see CCR5 expression on T cells in the tumour microenvironment and have mentioned this in the text on page 13 of the revised manuscript. We have never seen CCR1 expression on T cells and again, this is mentioned on page 13 of the revised manuscript.

The provided images lack of quantification and control for specificity. Considering that other cells than monocyte and macrophages might express the fluorescent reporters, it is unclear what the image really show.

As discussed above, and as mentioned on page 18 of the revised manuscript, using the reporter mice along with the recently developed CODEX technology will allow cell phenotype to be defined.

Renal tubules are very auto fluorescent and seems to be the main source of signal in Figure 5Bvii.

The renal tubules are indeed very autofluorescent and we apologise for the poor quality of this image. We have now used software to reduce the background autofluorescence and present this new image in revised Figure 5Bvii. For transparency we have also included the original image as new Figure 5—figure supplement 2 in the revised manuscript and discussed the use of the software on pages 11 and 25-26 of the revised manuscript. We hope that this enhanced image addresses the reviewer’s concern.

The authors correlate fluorescent reporter to the CK receptor expression using specific anti-CCR2, anti-CCR3 and anti-CCR5 on splenic cells to justify their conclusion of Figure 2. This claim needs reconsideration as the authors show only in the spleen, only on selected populations (Mo for CCR2, eosinophils for CCR3 and kidney Mac for CCR5) and only at steady state. Moreover, this claim is extended to CCR1 reporter for which protein expression cannot be validated.

The purpose of these experiments was to show, in exemplar cell types, that the antibodies faithfully recapitulate receptor expression. Cells chosen were selected on the basis of their known expression of the tested receptors. CCR2 and CCR3 are easily detectable in the spleen but CCR1 and CCR5 are not. These were therefore studied in kidney macrophages on which these receptors are expressed. For CCR1 our correlation is with cells expressing CCR1 transcript as revealed by in situ hybridisation. We have improved the wording of this section (pages 6-7) to clarify this point and hope that it is now acceptable to the reviewer.

The receptor /reporter correlation analysis already show that the reporter MFI does not strictly correlate with the antibody labeling MFI.

We hope that we have not misunderstood the point being made here by the reviewer. Basically, there is no reason to assume that the MFIs from the antibody and the reporters will be the same as they are on different fluorochromes. If what the reviewer means is that the MFI for the reporters doesn’t increase proportionately to antibody binding, then we would respectfully disagree and point out that it does increase proportionately and to a higher level (which is one of the advantages of using the reporters).

However, the authors make assumptions all along the manuscript that they can track the dynamic of CKR expression and draw mostly their conclusion based on the percentage of iCCR+ cells. The graphical abstract suggests that the CKR expressions per cell are modified in inflammatory conditions but does not reflect the change in cell frequency as supported by the data.

The reviewer is correct and throughout the manuscript we have referred to % iCCR reporter expressing cells and this indeed can be used to track dynamics of receptor expression but is limited in that we have not presented data on expression per cell. We have now added additional information on the MFI values in Figure 7 and MFIs are also present in terms of flow cytometry co-expression plots in Figure 4—figure supplement 1; Figure 5—figure supplement 1; Figure 8—figure supplement 1 and Figure 9—figure supplement 2. We hope that these additions are sufficient to address the reviewer’s concerns.

Overall more extensive characterization of this correlation in the different cell subsets for each receptor, tissues and experimental conditions need to be provided and the conclusions need to be redrawn accurately all across the manuscript.Beyond the points raised in the public review that need to be addressed I have a set of specific recommendations below:In the lung the gating strategy does not allow to capture non-classical monocytes which are more abundant than Ly6Chigh Mo and interstitial macrophages at steady state. Adding a CD43 staining would address that.

The reviewer is indeed correct that we have not evaluated expression on non-classical monocytes. We have now done this and include these data as Figure 4—figure supplement 4 in the revised manuscript and they are discussed on pages 10 and 14. These data show that, in contrast to the largely restricted expression of CCR2 on classical monocytes, these nonclassical monocytes express combinations of CCR1, CCR2 and CCR5.

This is a limit for the LPS model.Non-classical Mo should be expected to express CCR5 (from human studies at least). Is it confirmed by the multiple reporter mouse?

Please see the comment above

All across the manuscript the authors need to refer to the reporter expression and not the CCR protein whenever they use the fluorescence.

We fully agree with the reviewer and apologise for our carelessness in the original manuscript. We have altered this throughout.

Changes in % of iCCR+ cells do not reflect any down or upregulation of the receptor. For instance, line 336, the authors evaluate a 6-fold increase in CCR1 expression after IFNγ treatment. It seems that the frequency of Clover+ cells is increased which does not necessarily reflect increased CCR1 expression but accumulation of Clover+ cells or reduction of Clover- cells. Comparative MFI on iCCR+ subset in the different conditions along with the % of iCCR+ cells might reflect at least, change in the activation state of the cell. All statements regarding CCR expression should be carefully considered in the manuscript.

The reviewer makes an important point here. As discussed, we have now included MFI data for Figure 7 of the revised manuscript. These data demonstrate increased numbers of cells expressing Clover/CCR1, but not a significant increase in MFI (discussed on page 12 of the revised manuscript). We hope that inclusion of these data will satisfy the reviewer’s concerns.

Naming F4/80+ macrophages in the lung is unclear as AM are excluded from the analysis. The authors could consider using F4/80+ interstitial macrophage for clarification.

We have now added the word “interstitial” to the manuscript to avoid confusion and thank reviewer for this useful comment.

It appears that bimodal level of fluorescent reporter expression can be detected sometimes (CCR5, CCR1 in lung mac, CCR2 in Mo from air pouch membrane, CCR3 in air pouch eosinophils). It suggests that the authors deal with distinct subsets of cells which are considered as one. This need clarificationCCR2 MFI in kidney and lung Mo is bimodal (Figure 4-5Bvi). can the author explain why?

The reviewer is completely correct that we see this bimodal reporter expression for some reporters although this is really only for CCR2 and CCR3 reporters. This however is a fixation artefact (the lower intensity MFI is the artefactual one) and we have now highlighted this on page 12 of the revised manuscript. We apologise for not being clear on this issue in the original manuscript.

Dot plots showing co-expression of the reporters (as shown only FigS4Ci, ii) along with the pie charts would be more convincing and visual for the reader rather than the individual expression in all figures. Moreover, it will more clearly show potential spectral overlap between the different reporters.

We have now presented these data in Figure 4—figure supplement 1; Figure 5—figure supplement 1; Figure 8—figure supplement 1 and Figure 9—figure supplement 2 and they are discussed at appropriate points in the revised manuscript.

The assumption that Monocytes downregulate Ccr2 and upregulate Ccr5 and Ccr1 upon mac differentiation in vitro is fair. But this conclusion cannot be extended to the in vivo data as proposed. Adoptive transfer experiment of purified Mo should be performed to confirm this claim.

We have now included data from adoptively transferred CCR2+ monocytes showing that, in an inflammatory environment, they differentiate to macrophages that now express CCR1 and CCR5. These data are presented as Figure 4—figure supplement 3 and discussed on pages 9-10 and 16 of the revised manuscript.

Reviewer #2 (Recommendations for the authors):[…]Macrophages and DCs in the lung share many surface molecules but they can be distinguished by display of molecules such as F4/80 and CD88. On lines 264-265, the authors state that, "co-expression of CCR1 or CCR5 with CCR2 was more apparent in F480LoMHCIIHi than in F480MHCIIHi + IMs (Figure S4Ci-iv). I have two questions. First, in the latter cells, was the "+" supposed to refer to F4/80 levels? The authors seem to have accidentally left out the superscripted indicator of expression for that marker.

We thank the reviewer for pointing this out and, yes, the “+” refers to F480. We apologise for this carelessness and have now corrected this in the revised manuscript.

Second, why do the authors call the F480loMHChi cells 'interstitial macrophages', rather than conventional (c)DCs, specifically DC2? The data could be interpreted to mean that cDC2s maintain CCR2 longer than do macrophages, which would be an interesting observation.

We fully understand the reviewer’s concern here and the potential fine line between macrophages and dendritic cells. The macrophage nomenclature that we have used is based on a number of recent publications (Chakarov et al., 2019, Schyns et al., 2019 and Gibbings et al., 2017) in which these cells are classified as MHCII hi interstitial macs. We have now clarified this on page 8 of the revised manuscript.

Gibbings et al., 2017 DOI: 10.1165/rcmb.2016-0361OC

Chakarov et al., 2019 DOI: 10.1126/science.aau0964

Schyns et al., 2019 DOI: 10.1038/s41467-019-11843-0

There is no mention of lung CD103+ DCs (DC1), which can efficiently cross-present antigens to naïve T cells and have an important role is activation of CD8^+^ T cells. In addition to their display of CD103, DC1 can also be distinguished from DC2 by relatively low levels of CD11b. Thus, they can be identified as CD11chiMHCII hiCD103+CD11blow cells. However, I understand if the authors feel that this more detailed study is beyond the scope of the current paper.

We thank reviewer for this comment. The manuscript was designed to use exemplar cell types to demonstrate the utility of the reporter mouse strain and, as mentioned in response to comments from reviewer 1, we did not intend to include comprehensive leukocyte profiling. We are grateful for the reviewer’s understanding of the fact that this may be beyond the scope of current paper. We hope that the reviewer is happy with us not including these data in the revised manuscript.